



# Observations of Dust Particle Orientation with the SolPol direct sun polarimeter

Vasiliki Daskalopoulou[1,2], Panagiotis I. Raptis[3], Alexandra Tsekeri[2], Vassilis Amiridis[2], Stelios Kazadzis[4], Zbigniew Ulanowski[5,6], Vassilis Charmandaris[1,7], Konstantinos Tassis[1,7] and William Martin[8]

[1]Department of Physics, University of Crete, Heraklion GR-70013, Greece
[2]Institute for Astronomy, Astrophysics, Space Applications and Remote Sensing, National Observatory of Athens, Athens GR-15236, Greece
[3]Institute for Environmental Research and Sustainable Development, National Observatory of Athens, Athens GR-15236, Greece
[4]Physicalisch-Meteorologisches Observatorium Davos, World Radiation Center, Davos 7260, Switzerland
[5]Department of Earth and Environmental Sciences, University of Manchester, Manchester M13 9PL, UK
[6]British Antarctic Survey, NERC, Cambridge CB3 0ET, UK
[7]Institute of Astrophysics, Foundation for Research and Technology–Hellas, Heraklion GR-70013, Greece
[8]University of Hertfordshire, Centre for Atmospheric and Climate Physics Research, Hatfield AL109AB, UK

*Correspondence to*: Vasiliki Daskalopoulou (vdaskalop@noa.gr)

**Abstract.** Dust particles in lofted atmospheric layers may present a preferential orientation, which could be detected from the resulting dichroic extinction of the transmitted sunlight. The first indications were provided relatively recently on atmospheric dust layers using passive polarimetry, when astronomical starlight observations of known polarization were found to exhibit an excess in linear polarization, during desert dust events that reached the observational site. We revisit the previous observational methodology by targeting dichroic extinction of transmitted sunlight through extensive atmospheric dust layers utilizing a direct-Sun polarimeter, which is capable to continuously monitor the polarization of elevated aerosol layers. In this study, we present the unique observations from the Solar Polarimeter (SolPol) for different periods within two years, when the instrument was installed in the remote monitoring station of PANGEA - the PanHellenic Geophysical Observatory of Antikythera in Greece. SolPol records polarization, providing all four Stokes parameters, at a default wavelength band centred at 550 nm with a detection limit of $10^{-7}$.

We, overall, report on detected increasing trends of linear polarization, reaching up to 700 parts per million, when the instrument is targeting away from its zenith and direct sunlight propagates through dust concentrations over the observatory. This distinct behaviour is absent on measurements we acquire on days with lack of dust particle concentrations, and in general of low aerosol content. Moreover, we investigate the dependence of the degree of linear polarization to the layers' optical depth under various dust loads and solar zenith angles, and attempt to interpret these observations as an indication of dust particles being preferentially aligned in the Earth's atmosphere.

**Keywords:** dust polarization, polarimeters, dust orientation



## 1 Introduction

Mineral dust constitutes one of the most abundant atmospheric aerosols in terms of dry mass (Tegen et al., 1997) and, thus, plays a significant role to the radiative forcing of the global climate (Miller and Tegen 1998, Miller et al. 2014) being the dominant source of aerosol forcing downwind of major dust sources, such as the Sahara Desert (Li et al. 1996; Chaibou et al. 2020). In atmospheric transport models, dust particles are mainly assumed to be spherical which causes an overestimation of their gravitational settling speed leading, though, to a substantial underestimation of the coarse mode size distribution in the Earth's atmosphere (Adebiyi et al., 2022; Adebiyi and Kok, 2020). Furthermore, limitations using realistic irregular shapes for dust particles or their possible orientation to the analytical computation of the aerodynamic drag force in the gravitational settling schemes, usually leads to the assumption of randomly oriented particles or their volume equivalent spherical counterparts (Mallios et al., 2020). Although non-sphericity has been addressed previously (e.g. Ginoux 2003), only recent studies attempt to quantify its effect over larger particle sizes and take into account either particle orientation or asphericity to dust gravitational settling (Drakaki et al., 2022; Huang et al., 2021; Mallios et al., 2021). Similarly, in both active and passive remote sensing aerosol retrievals, particle shape is an important factor for modelling the scattering properties of dust particles (e.g. Dubovik et al., 2006; Gasteiger et al., 2011; Saito and Yang, 2021 and references therein).

Incoming solar radiation is considered unpolarized before it enters the Earth's atmosphere. Throughout its propagation in the atmosphere its polarization changes, through the absorption and scattering interactions with various atmospheric components including aerosol particles, water droplets, ice crystals and molecules. The transmitted (direct) sunlight is always unpolarised, except when it propagates through oriented particles in the atmosphere. Since dust grains are asymmetric and tend to be elongated, they preferentially scatter light polarised along their long axis. In this case, the extinction of the light polarization components is different (i.e., dichroic extinction) and the transmitted sunlight becomes polarized. The dichroic extinction of transmitted starlight through oriented dust particles is a well-documented phenomenon in the case of the interstellar medium, where dust particles are preferentially aligned along the galactic magnetic field lines. Dichroism measurements provide information on the magnetic field orientation, which is the dominant alignment mechanism there (Andersson et al., 2015; Dasgupta Ajou K., 1983; Kolokolova and Nagdimunov, 2014; Lazarian, 2007; Siebenmorgen, 2014; Skalidis and Tassis, 2020). Atmospheric dust may provide similar linear polarization (LP) signatures, as vertically oriented particles can lead to dichroic extinction of the transmitted sunlight. This was shown again for starlight observations, which included predominantly horizontally polarized light during a Saharan dust episode in La Palma (Bailey et al., 2008; Ulanowski et al., 2007). However, such measurements refer to column-integrated values that are not capable of determining the vertical distribution of the phenomenon throughout the dust layer. The latter is addressed by a novel polarization lidar for detecting dust orientation, nicknamed WALL-E, which is expected to provide valuable information for monitoring the phenomenon of dust polarization in the Earth's atmosphere (Tsekeri et al., 2021).

A potential mechanism that is capable of aligning the lofted dust particles, is the large scale atmospheric electric field that acts on charged dust particles through coupled electrical and aerodynamic torque interactions (Mallios et al. 2021, Ulanowski et al.



2007). For charged dust particles, the polarity of particle charge and the direction of the electric field influence particle motion and, therefore, the aerodynamic torque. Depending on its strength, the total electric field within the dust layer can: (a)

counteract the gravitational settling of large particles and (b) cause a preferential orientation of the non-spherical particles along the vertical direction. Mallios et al. 2021 stipulated that aerodynamic torque, caused by the misalignment between the center of pressure and center of gravity on a spheroid dust particle, tends to rotate it horizontally, while the electrical torque, caused by the misalignment of the electrical dipole moment and the electric field direction due to the asymmetry between the spheroid axes, tends to rotate the particle vertically. Particles of sizes less than 1 μm are always randomly oriented due to the

Brownian motion being dominant, while particles in the range of 1-100 μm can become vertically oriented for sufficiently large electric field strengths (Mallios et al., 2021). Although, enhanced electric field strengths attributed to charge separation within the elevated dust layers have been observed by both ground-based methods (Daskalopoulou et al., 2021a) and vertical profiling of the layer electrical properties (Daskalopoulou et al., 2021b; Nicoll et al., 2011), up till now, these studies display that the measured electric field strength may not be sufficient to overcome the threshold value that would orient particles

vertically. This hints that other factors, such as strong updrafts, can highly contribute to the phenomenon leaving the exact mechanisms of dust orientation a matter of intense discussion.

In the present work, we consider the recent gap in literature concerning the effect of oriented dust to direct sun polarimetric measurements and are motivated by previous observations of starlight dichroic extinction due to oriented dust (Bailey et al., 2008; Ulanowski et al., 2007), in order to provide consistent monitoring of dust orientation in the Earth's atmosphere. The

capabilities of direct sun polarimetric measurements are discussed in Kemp et al. (1987) and Kemp and Barbour (1981), setting the detection threshold as low as sensitivies of the order of $10^{-7}$, for perturbations to the polarization of forward scattered light. In order to investigate the above, we employ the direct sun solar polarimeter (SolPol) and report on signatures of dust particle orientation within the atmospheric column for two consecutive years of operation. SolPol is a passive ground-based polarimeter that has been developed at the University of Hertfordshire. It was initially implemented as a laboratory

experimental configuration in order to acquire both linear and circular polarization signatures of scattered light from biological materials and cyanobacteria, to demonstrate the detection capabilities of potential life presence on exoplanetary atmospheres (Martin et al., 2010, 2016). It was, after that, kindly conferred to the Remote sensing of Aerosols, Clouds and Trace gases (ReACT) team of the National Observatory of Athens for operation purposes that exceeded its indoor capabilities, as it was installed in the remote PANhellenic GEophysical observatory of Antikythera (PANGEA), under the framework of the D-

TECT European Research Council project. As we aimed at monitoring potential orientation signatures on-demand in elevated dust layers, the instrument was housed in an astronomical dome for protection and was operational during seasonal periods of intense dust circulation over the station area. During the overall period of operation in PANGEA, we conducted multiple test measurements with the new instrument setup to configure the best possible sequence and achieve accurate polarization measurements. We, also, determined the atmospheric conditions under which SolPol's detection limits were strained and

characterized the individual steps for the instrument's basic principle of operation. Detailed information concerning the



technical characteristics of the instrument are beyond the scope of this study and are subsequently provided in the form of an instrument manual and test measurements documentation in the Supplementary material.

In Sect. 2 of the paper, we present a comprehensive description of the instrument design, the side components that comprise the configuration and delineate the basic principles of operation focusing on measuring the Stokes parameters of the transmitted

light. In Sect. 3, we describe the process of acquiring quality-assured data with SolPol and the followed methodology for each measurement sequence regardless of the ambient conditions. Sect. 4 comprises the core observations section where we present the total acquired dataset for the selected two-year operation and compare between days affected by elevated dust layers and days without. Then, in Sect. 5 we discuss the dependence of the derived orientation trends to crucial parameters such as the Sun's position and the aerosol optical depth (AOD) so as to verify that the excess in LP is attributed to preferentially oriented

dust particles within the examined layers. Detailed calculations for the methodologies and technical descriptions presented in the paper are provided in the Supplement, as well as, in Appendix A.

## 2 SolPol: Components and principles of operation

### 2.1 Design

The design of the SolPol instrument is quite robust and follows that of its astronomical counterpart and predecessor, PlanetPol

(Bailey et al., 2008; Hough et al., 2006). Figure 1 illustrates the polarimeter individual parts, spanning from the head to the instrument support and data acquisition peripherals. A more detailed representation can be found in the SolPol manual/document under the Supplementary material of this paper. The instrument's principal operation is heavily based on a Hinds Instruments Photo Elastic Modulator (PEM) head (Figure 1), as most astronomical spectropolarimeters, due to the PEM high sensitivity to polarization fraction of the order of $10^{-6}$ (Hough et al., 2006; Kemp and Barbour, 1981; Povel, 1995). The

PEM is comprised of a birefringent crystal that is mechanically strained at a principal resonant frequency ($\omega$) of 47 kHz and induces a sinusoidal phase (modulation) retardation of $0.382*\lambda$ to the input signal, where $\lambda$ is the incoming light wavelength. By exploiting the fundamental vibration along the PEM crystal optical axis different polarization states are refracted in different directions, enabling the full Stokes parameters quantification with a single rotation of the PEM by 45° (Kemp and Barbour, 1981 and references therein).

The polarimeter has a default limiting field-of-view (FOV) aperture of 5.5 mm diameter with a possibility of employing different iris sizes. The default aperture exactly encompasses the solar disk with an apparent angular diameter of 0.5 degrees, provided that sun-tracking is stable throughout the measurement. This is ensured in our case before every observational sequence. As seen in the electronics schematic (Appendix A, Figure A1), there are no optical elements before the PEM, which is directly followed by a rotatable linear polarizer with its passing axis aligned to the PEM long axis per manufacturing. Both

the polarizer and PEM head can be rotated as a whole, through a Pyxis commercial rotator, and their position is controlled from a LabView virtual instrument program, which also controls the recording of data from the detector.





Neutral density (ND) filters are employed next, on the filter wheel (Figure 1), containing six RGB broad band ND filters with measured transmission curves and three 40 nm narrow band interference filters at 400, 550, and 700 nm centre wavelengths. The polarimeter capabilities are tuned at 550 nm monochromatic operation for the present study. The ND filter wheel is

followed by a simple Galilean telescope configuration, labelled as "Optics" in Figure 1, which collects and focuses the signal levels at the 1-cm silicon photodiode detector. The detector uses a transimpedance amplifier to generate the signal voltage. A 12-bit Analog-to-Digital converter (ADC) records the diode output signal and an SR830 Lock-in amplifier (labelled as "Peripherals") records the first ($1\omega$) and second harmonic ($2\omega$) AC de-modulation signals that are the outcome of the operation of the PEM. Lastly, SolPol continuously tracks the sun through a lightweight EQ3 SynScan astronomical mount. An external

CCD camera, also on the mount, assists with the visual alignment of the instrument. By way of clarification, we will refer to components (1) - (7) as the "assembly" for the remainder of this paper in order to better describe the required SolPol movement for the acquisition of the complete Stokes vector $[I\ Q\ U\ V]^T$ (e.g., Hansen and Travis, 1974).

SolPol is a non-imaging polarimeter, hence its calibration technique varies from most known imaging polarimeters (Povel, 1995). In principle, the linear polariser and the PEM are carefully mechanically aligned by careful nulling of the signal, with

respect to the polarizer rotation positions, from an arbitrarily polarised source. This sets the signal-to-noise ratio for the minimum detectable polarisation. Current instrument configuration has the capabilities of detecting a minimum of the order of $10^{-7}$. If the field-of-view with respect to the solar disk was asymmetrical, offset, (i.e., the alignment is not optimum) then the measured polarisations are expected to increase since the measurements will also include polarized diffuse sunlight. This makes stable closed loop Sun tracking a pre-requite for the minimization of measurement errors.

## 2.2 Principles of operation

SolPol measures the polarization fractions of linear polarization (expressed by the Q and U Stokes parameters) and circular polarization (V Stokes parameter) from the whole solar disk and the entirety of the atmospheric column, depending on its limiting field-of-view aperture and the choice of mounting telescope. The measurements of the degree of linear and circular polarization (i.e., $DOLP = \sqrt{Q^2 + U^2}/I$ and $DOCP = V/I$, respectively, e.g., general expression from Hansen and Travis

1974) have an absolute accuracy of 1% and precision of 1 part per million (ppm, $10^{-6}$) in polarization terms. The first observing sequence is performed with the assembly at the rest position (zero degrees) and the rotating of the linear polarizer in steps of 90 degrees. The corresponding relative positions of the PEM and linear polarizer for the assembly rotation is shown in Figure 2. For a complete polarizer rotation (360 degrees) we acquire measurements for four minutes. The specific sequence provides measurements of three of the four Stokes parameters. This is followed by the second observing sequence, performed after the

rotation of the entire assembly, about the PEM crystal optical axis, over 45 degrees. This observing sequence provides measurements of the fourth Stokes parameter (V), and measurements for the removal of the biases and residual polarizations due to high frequency strain of the PEM for another four minutes.





Therefore, each full measurement cycle has a duration of eight minutes in total and is comprised of two distinct sets: i. solar irradiance measurements on four positions of the linear polarizer at 41°, 131°, 221° and 311° with the assembly being at 0°,

this configuration provides measurements of the I, U and V Stokes parameters and ii. measurements for the same relative positions of the linear polarizer, but the whole assembly is being rotated by 45° which provides measurements of the I, Q and V Stokes parameters (Figure 2). Measuring I and V twice provides information for the removal of biases and residual polarization in the measurements.

The Stokes vector of light that reaches the PM can be expressed using the Mueller formulation (e.g., Van de Hulst, 1957),

considering as reference coordinate system the one of the incoming sunlight, as in (1):

$$\begin{cases} I'_{\alpha^o} = M_{Pol.,\alpha^o} M_{PEM} I \\ I'_{rot,\alpha^o} = M_{Pol.,\alpha^o} M_{PEM} I_{rot} \end{cases} \tag{1}$$

where $I = [I\ Q\ U\ V]^T$ is the Stokes vector of the input light polarization state, $I'_{\alpha^o}$ is the output polarization state with the assembly at zero degrees,

$$M_{Pol.,\alpha^o} = \frac{1}{2}\begin{bmatrix} 1 & \cos 2a & \sin 2a & 0 \\ \cos 2a & \cos^2 2a & \sin 2a \cos 2a & 0 \\ \sin 2a & \sin 2a \cos 2a & \sin^2 2a & 0 \\ 0 & 0 & 0 & 0 \end{bmatrix} \tag{2}$$

is the Mueller matrix of the linear polarizer at each position angle $\alpha° \in (41°, 131°, 221°, 311°)$ and

$$M_{PEM} = \begin{bmatrix} 1 & 0 & 0 & 0 \\ 0 & 1 & 0 & 0 \\ 0 & 0 & \cos \delta & \sin \delta \\ 0 & 0 & -\sin \delta & \cos \delta \end{bmatrix} \tag{3}$$

is the Mueller matrix of the PEM that induces an input sinusoidal retardation of

$\delta(t) = A\sin\omega t$, for A: the peak modulation amplitude and ω: the modulation frequency.

When the whole assembly is rotated by 45°, the reference coordinate system is rotated by the same angle, hence $I'_{rot,\alpha^o}$ is the Stokes vector of the output light for this rotational scheme. The incoming sunlight at each $\alpha°$ position, appears to be rotated by an angle of -45° and is calculated through the rotation matrix $R_{-45°}$ as $I_{rot} = [I\ U\ -Q\ V]^T$ (as in Freudenthaler, 2016, S.5.1.7; Martin et al., 2010; Supplementary material/SolPol manual, Eq. 8).

We utilize a Bessel functions expansion of the retardation δ and derive the measurements of the Stokes parameters of the incoming sunlight, at detector level, as a function of the linear polarizer angles and the assembly rotation (Table 1). The $J_n(A)$ are the n-order Bessel functions and, for the specific PEM, the modulation amplitude is fixed at A = 2.4048 so that $J_0(A) = 0$, which makes the Q- and U-dependent DC terms equal to zero. Third order and above harmonic frequency terms are considered negligible, $O$ (n > 2) → 0. We, then, sum over the four linear polarizer orientations per each assembly position, in order to

eliminate the $\cos 2a$ dependent terms on the I derivation shown in Table 1, while the marginal $\sin 2a$ residuals on the other Stokes parameters are also accounted for in the data processing chain.



As discussed in Bailey et al. (2008) and Ulanowski et al. (2007a), in case of aligned particles in the atmosphere, DOLP is expected to increase over larger solar zenith angles (SZA) since the direct sunlight travels through a larger airmass, and the effect of the dichroic extinction increases. With the increase of the SZA, the particle alignment angle changes with respect to the direction of observation, which is expected to influence the dichroic extinction and the measured linear polarization.

## 3 Data and Methodology

### 3.1 Data acquisition

SolPol has been primarily installed at the PANhellenic GEophysical observatory of Antikythera (PANGEA) in the remote island of Antikythera (35.861° N, 23.310° E, 193 m asl-above sea level) since September 2018. The island covers an area of just 20.43 km² and is devoid of heavy human activity which makes it an ideal background observatory for the Eastern Mediterranean. The station location is carefully selected, as the island is placed at a "crossroad" of different aerosol air masses (Lelieveld et al., 2002), with NNE winds being prominent between August and February, while in spring and early summer the western airflows largely favor dust transport from the Sahara desert. The prevailing meteorological conditions are representative of the broader region with warm and dry days in the summer, while colder and wetter days are typical during the winter. Weather limitations and increased salinity due to the coastal site effect, led to housing the SolPol in a specifically built motorized astronomical dome that offered protection from various extreme weather conditions and where it could be mounted to the EQ3 tracker for on-demand direct sun observations.

The datasets presented in this study were acquired from May 2020 to June 2021, under various atmospheric conditions and aerosol loads, specifically targeting days with dust outbreaks but also days that are representative of the local background conditions. Even a small cloud coverage can highly influence the transmitted light polarization state on optical wavelengths due to multiple scattering from cloud droplets or due to dichroic extinction from aligned ice crystals, in the case of cirrus clouds, that can lead to polarization effects in the forward direction (Baran, 2004; Bohren, 1987; Emde et al., 2004; Ulanowski et al., 2006). Accordingly, the affected SolPol measurements are screened and in cases of prolonged transient clouds over the station the instrument was shut down. All the measurements are always direct sun.

### 3.2 Processing chain

The raw data acquisition comprises a set of five similar measurements over each polarizer angular position per PEM rotation, with each measurement associated with auxiliary information such as the polarizer specific angle, the PEM position, measurement time in UTC, the voltage outputs in the ADC and the two channels of the lock-in amplifier. In order to optimize data handling, we perform an initial data reduction to each measurement sequence by selecting the fifth measurement set for the same linear polarizer positions.

The DC signal amplitude returned from the PM detector is simultaneously recorded by the ADC in volts (Appendix A Figure A1) and gives the magnitude of the total irradiance I ((4) as seen in Table 1. By turning the linear polarizer at specific positions,



we acquire the amplitude and phase of the rms (root-mean-square) AC modulated signals as outputs of the lock-in amplifier respective channels, and each is directly proportional to the magnitude of Q and U for even $\omega t$ ((5 and (6) and to the magnitude

of V for odd $\omega t$ ((7). However, the two signals are processed through separate electronic channels on the amplifier, inducing a different proportionality factor for linear and circular polarization that depends on the modulation frequency and which is considered in data processing. Therefore, the recorded voltages from which we derive the Stokes parameters, are:

$$v_{DC} = I/2 \tag{4}$$

$$v_{2\omega,0°} = U J_2(A)/\sqrt{2} \tag{5}$$

$$v_{2\omega,45°} = -Q J_2(A)/\sqrt{2} \tag{6}$$

$$v_{1\omega} = V J_1(A)/\sqrt{2} \tag{7}$$

where $J_1(A) = 0.5192$ and $J_2(A) = 0.4317$ are the first and second order Bessel functions for A = 2.4048.

For data smoothing, we impose the Chauvenet criterion on each daily dataset with a 120-min duration, which is the maximum

consecutive measurement duration of SolPol, and detect outliers to the Stokes parameters due to instrument misalignment and persisting perturbations to the measurements caused by passing low clouds. The criterion is applied once with the relation below:

$$N\, erfc\left(\frac{d_i}{\sqrt{2}s}\right) < \frac{1}{2} \tag{8}$$

for a deviation of $d_i = |S_i - \bar{S}|$

where $\bar{S} = \frac{1}{N}\sum_{i=1}^{N} S_i$ is the mean output of Q, U or DOLP from the processing algorithm, $S_i$ is the $i^{th}$ output for $i = 0,.. N$ and

$N$ is the total number of computed triplets for each two-hour subset. Lastly, erfc(x) is the complementary error function[1] and $s$ is the unbiased sample variance[2].

### 3.3 Polly[XT] lidar measurements

For the comprehensive characterization of the vertical distribution of aerosol optical properties, we exploit the profiling capabilities of the co-located Polly[XT] Raman polarization lidar (Engelmann et al., 2016) of the National Observatory of Athens

(NOA),          as          part          of          the          European          Aerosol          Research          Lidar          Network          (EARLINET     -

---

[1] $erfc(x) = 1 - erf(x) = 1 - \frac{2}{\sqrt{\pi}}\int_0^x e^{-t^2} dt$

[2] $s^2 = \frac{\sum_{i=1}^{N}(S_i - \bar{S})^2}{N-1}$



https://polly.tropos.de/calendar/location/38?&individual_page=2020, last visited: 16/12/2022). This multi-wavelength system is equipped with three elastic channels at 355, 532 and 1064 nm, two vibrational Raman channels at 387 and 607 nm, two channels for the detection of the cross-polarized backscattered signal at 355 and 532 nm, and one water vapour channel at 407 nm. The basic lidar quantity used is the Volume Linear Depolarization Ratio (VLDR, $\delta_v$) at 532 nm. VLDR (%) is the ratio of

the cross-polarized to the co-polarized backscattered signal (Freudenthaler et al., 2009), where cross- and co- are defined with respect to the plane of polarization of the emitted laser pulses. VLDR values are influenced by both atmospheric particles and molecules, with high $\delta_v$ values being indicative of irregularly shaped particles (i.e., atmospheric dust). Typical $\delta_v$ values for Saharan dust are in the range of 20% to 30% at 532 nm (Haarig et al., 2017). The 2020 and 2021 dust events reaching Antikythera, presented herein, are characterized by generally large concentrations of airborne dust particles from the middle

of the day onwards with $\delta_v$ exceeding 15%, followed by dust settling towards the ground at late afternoon. The dust layers are mostly detached and homogeneous with some vertical mixing with non-depolarizing marine particles occurring within the Marine Boundary Layer (MBL).

### 3.4 Ancillary aerosol information

The Aerosol Optical Depth (AOD) was monitored by the CIMEL sunphotometer, part of the NASA Aerosol Robotic Network
(AERONET - https://aeronet.gsfc.nasa.gov/, last visited: 12/12/2022), which was also installed in PANGEA along with Polly[XT] and SolPol. For all the cases examined in this paper, mean AOD values were used and varied from 0.047 to 0.502 at 500 nm. Moreover, near-surface wind speed measurements presented here were obtained from a co-located Davis Vantage Pro2 weather station at an altitude of 198 m asl.

### 3.5 Mean dark bias correction

For the majority of the cases, bias readings (e.g., PM detector temperature-dependent offsets) provided by dark signal measurements, are interspersed between all sequences for the quantification of biases on the measured DC signal. Detailed dark tests carried out with SolPol can be found in the Dark/Tests document under the Supplementary material of this paper. Bias readings are not treated as regular polarization measurements but instead the dark flux is solely exploited as an absolute value. The mean dark DC flux, $I_{mean,dark}$, is one order of magnitude lower than the total intensity measurements $I$ for the

cases that it maximizes, while there are cases that it becomes less than two orders of magnitude. The days that lack such measurements are treated instead with the mean dark flux of all the days during which dark measurements were acquired. In order to acquire representative background conditions over the station, we selected a "clean day" with small non-depolarizing aerosol concentrations and layer homogeneity, no clouds and a relatively smooth near-surface wind profile. During relatively calm sea surface conditions, specular reflection phenomena intensify and could potentially contaminate the direct-sun

measurements with linearly polarized reflected light of the same magnitude as the expected dichroic extinction from the dust particles. We do not observe such a behavior with SolPol, although the observational site is in close proximity to the wavy sea surface (about 1.5 km in optical path towards the eastern direction). This can be explained as the established viewing





geometry, which is confined by low elevation and rapidly increasing SZA, does not enclose Brewster angle reflections within the instrument field-of-view (FOV).

In Figure 3, we examine the change in the DOLP calculations, for a specific case study, to the subtraction from I of one of the three following parameters: i. the minimum recorded dark intensity for the specific day, $I_{min,dark}$, ii. The average dark intensity, $I_{mean,dark}$ and iii. The maximum value of the recorded dark intensity, $I_{max,dark}$, as a way to constraint the sensitivity of DOLP to the dark current variability. Figure 3a, shows the four values derived for DOLP and their variability within the day, considering different values for the dark intensity. The measurement spans from 06:00 to 17:00 UTC and DOLP is plotted

in ppm reaching maximum values of ~35 ppm for every case, as seen in the zoomed representation of the measurement (Figure 3b). The relative difference between the methods is below 0.1%, which is considered negligible, and subsequently we keep the subtraction of $I_{mean,dark}$ for the remainder of our processing chain. The polarimeter line-of-sight is unobscured during the observing sequence, as depicted to the typical and unperturbed behaviour of the measured irradiance (Figure 3c). Different colours signify the corresponding measurement subsets within the day and the recorded wind speeds for 30-minute averages

do not exceed 4 m/s (Figure 3d). For the same bias correction methodology under a dust-affected day, Figure A2 and Table A1 of the Appendix A showcase the similarities with the clean day case and the measured dark DC fluxes.

## 4. Observations

In this section, we present the observational dataset of SolPol acquired in 2020 and 2021, when the instrument was installed at the PANGEA observatory in Antikythera, focusing on linear polarization measurements. SolPol's behaviour during this

period is consistent with previous operations when tested in indoors conditions. Generally, observations were initiated just after sunrise and terminated before sunset when the Sun disk started to be shadowed by nearby topography. Before each deployment, the instrument general condition was thoroughly checked in terms of instrument alignment, tracking accuracy, power supply levels consistency and the optimal communication between peripherals and the assembly control units to ensure minimum mishaps or potential disconnections of the assembly rotator.

Circular polarization (CP) by scattering at aerosols, cloud droplets, and ice crystals is several orders of magnitude smaller than linear polarization, while scattering by the molecular atmosphere is expected to not cause any circular polarization (e.g. Emde et al., 2017). This is similar to what is observed by SolPol in the forward scattering direction, where CP is 1-2 orders of magnitude less than LP under dust presence and near-zero for background days. A thorough review by Gassó et al. (2022) on circular polarization by atmospheric constituents, proposes that circular dichroism from aligned atmospheric dust particles

could produce CP features but its magnitude has not been defined yet. Since the phenomenon is expected to be borderline detectable with the discussed polarimeter configurations, in interstellar dust degrees of LP and CP by aligned grains become comparable (Kolokolova and Nagdimunov, 2014), we choose to focus on linear dichroism observations and attempt to reproduce the findings of Bailey et al. (2008). For that reason, the Stokes V parameter observations and analysis with SolPol are not discussed here but we aim to further process these data in future studies.





## 4.1 Polarization measurements

Figure 4 consists of a collection of all the recorded days within this timeframe, where measurement set durations within the day varied between consecutive sixteen-minute sequences to the maximum two-hour retrievals. We subplot the Q/I (black dots), U/I (purple dots) and DOLP (blue dots) parameters again in ppm with time of the day in UTC, for cases of both lofted dust layers reaching the station and cases with background/clean conditions.

We label the days as dust driven (D) days with various dust loads, clean days (C) that are mostly characterized by concentrations of marine particles within the MBL, but lack dust particle presence, and of half-clean (HC) days where dust presence becomes prominent sometime within the day and potentially mixes within the MBL. This classification is inferred from the Polly$^{XT}$ VLDR values for each day (not presented here for all the cases for brevity reasons, but can be publicly accessed in the Polly-NET website https://polly.tropos.de/) and the AOD values by AERONET. We also include a dedicated test day (TD) with diffuse light testing for various SolPol aperture sizes (Section 5.3) and two consecutive half-day retrievals with a relatively high dust load for which a larger aperture was used.

### 4.1.1 Reference days

Focusing on DOLP behaviour for clean days, the measured polarization values span from 0.5 x 10$^{-6}$ usually at local noon when linear polarization of transmitted sunlight is expected to minimize in near zenith viewing angles, up to 50 x 10$^{-6}$ in large solar zenith angles (i.e., early in the day or in the afternoon). Minimum linear polarization values for these days are consistent with early observations of polarization from the whole Sun at the same spectral region as SolPol, noted in Kemp et al. (1987) work using a similar polarimeter apparatus. Moreover, the maximum values indicate the maximum bias consisting of pure Rayleigh scattering events, by the molecular atmosphere at low altitudes, within the SolPol FOV and the total instrumental errors that can add to the light polarization state. A statistical analysis of all the clean day results is discussed in detail in Section 5.1 and determines the exact reference instrumental bias.

A typical measurement for a clean day behaviour is provided for the August 29$^{th}$ 2020 case study which was chosen above for the bias correction comparison and is delineated in Figure 5. The top panel presents the daily progression of the linear polarization parameters in ppm expressed through the Q/I (black dots), and U/I (purple dots) Stokes parameters and DOLP (blue dots). Measured linear polarization does not exceed the 50 ppm threshold regardless of the instrument viewing angle from the zenith, the average daily AOD has a low value of 0.049 in 500 nm, while the SolPol retrieval is compared to the time-height plot of the VLDR at 532 nm, as retrieved from the Polly$^{XT}$ lidar. Low $\delta_v$ values (<5%), shown in blue tones, denote that there is a generally low concentration of non-depolarizing particles within the MBL for the total duration of the SolPol measurement, thus no indication of desert dust presence.





### 4.1.2 Days with dust events

Under dust driven days, the dissimilarity becomes apparent. The most interesting feature with the presence of dust particles are the large polarization values towards the morning and evening of each measurement day. Linear polarization values reach up to 700 ppm in extreme cases. Such values are exhibited during heavily dust affected days, e. g. from the 16/05/2020 to 18/05/2020 where we have recorded the largest dust optical depth values for the specific period of observation (Figure 4). The increasing trend of the DOLP values with time (usually after 14:00 UTC) is mirrored in early morning measurements as the

instrument monitors the atmospheric column with a decreasing SZA. This is our first consistent indication that preferentially - vertically or horizontally - aligned dust particles could be present in the observed layers and cause the dichroic extinction of transmitted sunlight through the layer. The specific trend is also in line with the reported observations in La Palma by Ulanowski et al. (2007), which were later demonstrated in detail by Bayley et al. (2008), that showed a fractional linear polarization increase with increasing SZA during dust events. And that polarization exhibited changes for different days

roughly in proportion to the change in dust optical depth. Figure 4 illustrates the collective and quantitative behavior of the observed days for up to 200 ppm so that the clean day trends are also discernible in comparison, while the exact polarization values are discussed in the following sections where we group all the above observations with respect to the solar zenith angle. To better describe these high values of the dichroic extinction of sunlight passing through dust layer(s), a dust-affected case study on September 2$^{nd}$ 2020, is also highlighted (Figure 6). During the specific day, a transient dust layer originating from the

Western Sahara travelled towards the station, and was monitored reaching the PollyXT lidar system already one day before, at altitudes between 2.5 km and 5 km. The dust layer continued progressing through the day with some downward mixing of settling dust particles within the MBL occurring between 04:00 and 06:00 UTC, but then the layer detached and the larger concentrations, denoted with red tones in the VLDR quicklooks (> 20-25%), accumulated between 2 km and 3 km creating a thin layer that persisted within the day. It is also observed that the near-ground dust concentration is very low, with the very

thin layer created below 500 m after about 15:00 UTC, being potentially a mixture of dust particles and particles of marine origin with $\delta_v$ values around 15% (at this height range lidar overlap issues may be present). The day had zero cloud cover and a nominal irradiance curve was recorded by the SolPol detector (Figure A2). As presented in the polarization graph (Figure 6, top panel), the degree of linear polarization shows a decreasing trend from early morning as the day progresses towards local noon. Although the measurement sequence started a bit later than the previously presented clean day sequence, the trend is

clear. The DOLP values are low, ~2 ppm near zenith, and reach up to 400 ppm when the observed increasing trend maximizes at the end of the sequence just before 16:00 UTC. The excess in linear polarization above the 50 ppm threshold (inferred from the reference day) becomes prominent for this case.

### 5. Analysis and discussion

In this section, the basic factors that can influence the linear polarization behaviour and produce the excess polarization trend

are discussed. We, firstly, analyse the dependence of DOLP to the SZA and determine the instrumental noise level based on





our reference days. Secondly, we illustrate the evolution of DOLP with respect to the increasing optical depth and airmass. We further attempt a linear regression between DOLP and high AOD values, while lastly the effect of incident diffuse light, over SolPol's aperture, to the polarization trend is discussed for a dedicated test measurement day.

## 5.1 DOLP dependence on SZA

As the SZA increases, particles that are preferentially aligned can be viewed by different angles and in the aspect that larger number concentrations are present within the instrument line-of-sight. It is, therefore, expected that on near-zenith angles, particles that are aligned with their long axis vertically will not influence the polarization and result in near zero values, similar to what was previously reported on aligned grains (Bailey et al., 2008; Kolokolova and Nagdimunov, 2014). In Figure 7, we present an analysis of DOLP measurements in ppm as a function of the solar zenith angle in degrees, for the cases of the

selected clean/reference days (Figure 7a) and under the observed dust events (Figure 7b). A total of eight days per category were used, while days labelled as half-days, the test day and two days with larger irises were excluded. LP in clean days does not exceed the strict 50 ppm limit that was discussed for the single case of Section 4.1.1, regardless of the SZA. This is clearly depicted in the zoomed representation (Figure 7c) that modifies the instrument noise threshold (shaded area) accordingly, given that the dataset is statistically significant and that clean days are interspersed between the recorded dust events.

Figure 7b illustrates that LP increases with solar zenith angle under dust driven days, a trend that starts at about 30° and is obvious mainly after 50° SZA, with a maximum value of $\sim 7 \times 10^{-4}$ being observed at 74°, which is over one order of magnitude larger than the noise level. Moreover, the polarization curve changes from day to day roughly in proportion to the change in the dust optical depth. This becomes more apparent for large SZAs when dust particle concentrations persist within the day. The segregation between reference days and dust driven days, along with the distinct behaviour of DOLP with increasing SZA

is consistent with the findings from stellar polarimetry (Bailey et al., 2008; Ulanowski et al., 2007) and provides the first indications of particle preferential orientation with a solar polarimeter.

### 5.1.1 Biases

The observed DOLP variation from clean-to-clean day (Figure 7c) represents the induced instrumental error under individual measurements, as the differences would be predominantly attributed to the temporal variability of the molecular atmosphere

and the dependence of direct solar irradiance to the recorded AOD (e.g. Gueymard, 2012). The threshold value of 50 ppm, is also orders of magnitude larger than other systematic errors in linear polarization caused, for example, by a slight misalignment of the PEM surface with the incoming light axis. According to Kemp et al. 1981 this is of the order of $10^{-7}$ for misalignment angles of 0.1 degrees which is way beyond any thermal crystal expansion effects or external mechanical stresses to the PEM head in the installation site. Tracking offsets that gradually result to slight angular deviations, cause the targeting of larger off-

disk areas and can be a factor that contributes to the characterized instrument bias, as linear polarization increases near the solar limb (e.g. Stenflo, 2005). Although, we performed a test for identifying the effect of diffuse light in the recorded linear polarization signal through incremental alterations of the aperture size (see Section 5.3), sky-scanning tests with a few degrees





offset (on steps of 0.5°) on the four cardinal orientations were heavily limited by the PM detection capabilities and were not conclusive.

### 5.1.2 Polarization angle

Information on particle alignment can be extracted from the polarization angle $\chi$ (Electric Vector Polarization Angle, EVPA). EVPA is the angle between the plane of light polarization and the plane of reference, i.e. the perpendicular plane to the optical axis interpreted by the PEM photoelastic crystal. The formula for the calculation of the EVPA is described in the Appendix (9) and Figure A3 shows the SolPol viewing geometry with respect to the SZA and the calculated EVPAs, as the polarization angles that correspond to measurements of DOLP that are larger than the characterized noise threshold of 50 ppm, which are essentially the observed dust cases. In order to have significant linear polarization and assuming that the particle is elongated, then the particle long axis should be perpendicular to the major axis of the polarization ellipse (Bailey et al., 2008), shown in Figure A3a, as the parallel polarization component is predominantly scattered by the particle while the perpendicular one is transmitted by the particle. The dust cases, here, present a distribution of EVPAs, centred mainly at the values of ~20°, 70° and 150° regardless of the increasing SZA (Figure A3). This points towards the particles being oriented at specific angles such that, preferentially vertically aligned particles correspond to ~110° (EVPA 20° ± 90°) mean orientation angle and particles that are preferentially horizontally aligned at mean angles of ~160° (EVPA 70° ± 90°). The observed linear polarization increase at large zenith angles implies that particles adopt both preferential orientations, considering the temporal variation of the dust layers for our observational dataset. Moreover, the increasing trend can be expected to be steeper for vertical particle orientation, because the transition from the zenith (either no dominant orientation or polarization), towards the horizon is stronger for vertical orientation as opposed to horizontal orientation. At horizon level, vertical alignment appears total, while horizontal alignment is only partial due to the fact that particles are still randomly oriented in the horizontal plane.

By the EVPAs considered in this study (Fig. A3b), particles tend to mostly adopt the horizontal alignment, which is in accordance with previous particle orientation studies stating that the hydrodynamic forces dominate the alignment, as opposed to the electrical forces, and tend to orient the particle horizontally as it falls within the atmosphere (e.g. Mallios et al., 2021, 2022 and reference therin). Nonetheless more synergistic studies on particle dynamics that will take into account the full particle scattering matrix, e.g. through the WALL-E lidar observations (Tsekeri et al., 2021), but also microphysical parameters such as, the particle size distribution, particle charge and asphericity, will be conducted in order to verify our findings and potentially explain the dominant mechanism that could reproduce such an orientation signature.

### 5.2 DOLP dependence on AOD

In this section, we examine the correlation of the observed excess polarization with the optical thickness of the dust layer. We expect the correlation because dichroic polarization is an extensive property of the particles, hence, its strength increases with the number of particles present in the incoming light path. As shown in the Figure 8 comparison, the DOLP measurements are given for different aerosol optical depth values with respect to the SZA (Figure 8a) - or the airmass (i.e. 1/cosSZA) -





respectively (Figure 8b), for all the acquired cases except the half-clean days and the dedicated test day. We are using the daily average AOD values per full day of SolPol measurements and in the case of days with sparse polarization measurements (either in the morning or afternoon), we derive the mean AOD for the timeperiod that is closer to the measurements. Star markers denote the dust cases (D), circle markers are used to distinguish the clean days (C) and the colour scale referes to various AOD ranges. The maximum recorded AOD for the duration of our measurements in PANGEA observatory was 0.5 at 550 nm, which

is large for the characteristic transport conditions of eastern Mediterranean, but relatively small compared to the massive transported loads that are monitored on near dust sources.

We expect that for small AOD values (< 0.1) the linear polarization values will be within the noise threshold as derived from the clean days behavior, as is the case here and since there are no observed dust events with such small loads there are no increasing trends exhibited (dark blue circles). As the optical depth increases, we observe the linear polarization upward feature

for a specific SZA, mainly above 40°, intensifying in proportion to the AOD with the highest values of DOLP recorded under heavier dust loads. This indicates that dichroic extinction is potentially enhanced due to the presence of a larger concentration of preferentially oriented dust particles under fixed viewing angles and, consequently, for a stable airmass, as seen in the linear relation of Figure 8b. The observed excess in LP is linearly proportional to the increasing airmass, which could be attributed to the particle viewing geometry for fixed AODs or the amount of aligned particles for a fixed viewing geometry as DOLP

increases more rapidly for days with larger AODs (Figure 8b). Moreover, this is shown when we inspect the corelation between DOLP and the AOD corrected with airmass (slant optical depth) in Figure 9. The distinction has a threshold for AODs above 0.1 that correspond solely to the presence of dust and as we move to larger loads this embeds an increase to the linear polarization values above the noise threshold for even smaller airmasses. The corelation strength could be tested and reproduced in future studies with dust loads close to one, as the potential particle orientation could affect the geometric

formalism for the derivation of the AOD.

We should also note though that the increase of DOLP values with AOD may as well indicate contamination of diffuse light (and corresponding polarization properties) in our direct-sun measurements. Thus, it may as well be that we do not measure (only or at all) dichroic extinction, but linear polarization of the diffuse light, with the contamination being stronger for larger AODs. For this purpose we have performed a test for the influence of the diffuse light in our measurements (Sect. 5.3), and as

a future step we plan to perfrom a more complete test, by acquiring observations of high-AOD pollution cases, where we know that the particles are spherical and thus produce no dichroic extinction.

**5.3 Diffuse light contribution**

In order to check whether there is a significant contribution to the linear polarization signal attributed to the incoming diffuse light in the polarimeter, we have performed a full day of tests with alternating iris sizes, under a dust event in June 2021 that

reached the PANGEA station (Supplementary material, SolPol manual). When increasing the aperture size, we increase the measured diffuse light and subsequently the expected contribution to the observed linear polarization will become prominent on the larger iris sizes. During June 25th 2021 measurements, large dust concentrations were present within the day with a



homogeneous layer at altitudes up to 5 km and the optical depth measured by the CIMEL sun-photometer was relatively stable with a mean value of 0.370 at 500 nm. The test procedure comprised alternating iris sizes from small (at 4.5 mm), regular (at 5.5 mm, i.e., the one used for regular SolPol measurements) to large (at 7 mm) through 16-minute measuring intervals so as to ensure that ambient conditions are relatively stable and polarization does not change significantly with the instrument viewing angle (Figure 10:Figure 10). The 4.5 mm iris was chosen with respect to a threshold voltage signal to the detector of 1.2 V at large zenith angles. The measuring sequence is consistent with tight observational intervals up until the local noon, but then data become sparse due to SolPol peripheral communication failures. All the dark measurements were performed after each triplet with a closed lid, closed dome and a normally tracking mount.

As seen in Figure 10, the measured DOLP (black dots) follows a generally decreasing trend from early morning to noon, and a generally increasing trend from noon to afternoon, due to the decreasing and increasing SZA, respectively. This behaviour is disrupted when we change the size of the iris for consecutive measurements until noon. When the iris size changes from 7 mm to 4.5 mm on morning to noon measurements, DOLP increases instead of the expected decrease due to less diffuse light being received by the detector. This proves that by increasing the aperture size, DOLP is less than what was initially expected thus diffused light does not contribute to the polarization trend. Therefore, the measured linear polarization can be attributed solely to the transmitted sunlight dichroic extinction rather than a diffuse contribution.

## 6. Conclusions

We report on extensive linear polarization measurements with a direct sun polarimeter, the SolPol, and detect signatures of dichroic extinction of sunlight in cases of transient dust layers above the PANGEA observatory, in Antikythera island, Greece. This phenomenon could be attributed to potentially preferentially aligned dust particles within the instrument line of sight towards the sun, a concept that reproduces earlier measurements of excess linear polarization of starlight through observations of astronomical sources with known polarization signatures (Bailey et al., 2008; Ulanowski et al., 2007). The similar behaviour, found here, is expressed as a steep increase to the measured DOLP when the Sun is far from its zenith position and when significant dust particle load is present in the atmosphere. The increasing trend is not found under conditions that are characteristic of the background aerosol concentrations in the area. We describe the characterization of the instrument capabilities and quantify the measurement bias from the background reference and through dedicated testing days. In addition, we monitor the excess polarization over various dust loads where the linear polarization feature is persistent and extract the dependence on the AOD. We find that as the AOD increases, the linear polarization values become larger overall and, therefore, the increasing trend intensifies for viewing angles closer to the horizon, which is again consistent with the previously highlighted work.

Out of the total 24 days of SolPol observations, pronounced increase of DOLP up to 700 ppms only in the (lidar characterized) dust days have been observed as a function of the AOD (sun photometer measured) and of the air mass. Investigating the polarization angle for most of these days, horizontally and vertically oriented particles can explain both DOLP increase with





solar zenith angle and also the EVPA levels. However, it would require more measurements with well characterized aerosol optical properties to try to generalize on the capabilities of ground based polarimeters to accurately estimate dust particle orientation. Further work is intended to test the SolPol response on aerosol layers that are not likely to exhibit particle orientation due to the shape of the particles, such as in pollution or smoke layers, and are expected to not affect the polarization of direct light. The anticipated vertically resolved measurements of the novel WALL-E polarization lidar, jointly with the

SolPol orientation signatures, can provide strong proof of the existence of oriented aerosol particles in the atmospheric column and elucidate the nature of the alignment process. Our observations have multiple implications for the parametrization of natural dust in aerosol models and for radiative transfer calculations, as particle orientation affects scattering properties. More complementary studies are encouraged as they can pave the way for new aerosol remote sensing developments.

**Appendix A**

Figure A1 (here)

Table A1 (here)

Figure A2 (here)

**Polarization angle distribution with SZA**

Electric Vector Polarization Angle (EVPA) indicates the angle between the plane of polarization and the plane of reference

and is defined as (e.g., general expression from Hansen and Travis 1974):

$$\chi = \frac{1}{2}\tan^{-1}\left(\frac{U}{Q}\right) \; with \; 0 \le \chi < \pi \qquad (9)$$

We are using the normalized U/I and Q/I values as the Stokes input parameters and take into account the constraint that the $\cos 2\chi$ of angles differing by $\pi/2$ should have the same sign as Q/I. In Figure A3, we present the EVPA in degrees with respect to SZA for the total SolPol measurement days. The EVPAs are separated in two categories, those that correspond to DOLP values less than 50 ppm (noise threshold) and the ones for DOLP values larger than 50 ppm. If the particles within the light

path are oriented with their long axis perpendicular to the major axis of the polarization ellipse, then the mean angle of particle orientation should be 60°, which means that particles are predominantly randomly oriented. When examining the EVPAs that correspond to DOLP values greater than 50 ppm (Figure A3), there is a clear distribution of angles in three clusters at ~20°, 70° and 150° that remain centred around these angles regardless of the increasing SZA. The linear feature starts for SZAs between 30° to 60° which is expected as some dust days exhibit dichroic extinction traits relatively close to zenith viewing

angles. The 20° and 70° polarization angles denote particles being preferentially aligned to an angle of 110°, almost vertically aligned (yellow tab, Fig. A3) and to 160° which correspond to being predominantly horizontally aligned (green tab, Fig. A3) and to 60° (red tab, Fig. A3).



Figure A3 (here)


**Authors Contribution:** VD installed and operated SolPol in all locations, collected and processed the data, provided physical interpretation to the measurements, authored the instrument manual, prepared the data repositories and the paper with contributions from all co-authors. VD and PR tested and optimized the instrument and acquired the measurements shown

herein, while AT and VA formulated the measurement strategy. AT also offered her expertise in dust particle orientation for the key physical interpretation of the measurements and manual preparation. VA directed the preparation of the paper, supervised the study, offered his specialty in lidar data interpretation and gave insight to the linear polarization measurements. SK helped in the realization of the measurement strategy and the paper preparation, offered his expertise in solar radiation measurements and provided valuable input to the measurement interpretation. JU offered his expertise in dust polarimetry and

unique conceptualization of the phenomena and provided scientific consultation through the entire process. VC provided insightful comments concerning dust orientation signatures. KT provided valuable scientific consultation concerning the acquisition process and the instrument operation as an expert to dust polarimetric observations. Lastly, WM kindly conferred the instrument, co-supervised the study and provided scientific consultation on data outputs and the processing chain.

**Acknowledgments:** This research was supported by data and services obtained from the PANhellenic GEophysical Observatory of Antikythera (PANGEA) of NOA. The authors would like to acknowledge support of this work by the project "PANhellenic infrastructure for Atmospheric Composition and climatE change" (MIS 5021516) which is implemented under the Action "Reinforcement of the Research and Innovation Infrastructure", funded by the Operational Programme "Competitiveness, Entrepreneurship and Innovation" (NSRF 2014-2020) and co-financed by Greece and the European Union

(European Regional Development Fund). We are grateful to EARLINET (https://www.earlinet.org/) and ACTRIS (https://www.actris.eu) for the data collection, calibration, processing and dissemination. VD would like to thank all the NOA-ReACT members for their help and support during the data retrieval process.

**Financial Support:** This research was supported by D-TECT (Grant Agreement 725698) funded by the European Research Council (ERC) under the European Union's Horizon 2020 research and innovation programme. VD would like to acknowledge that this research is also co-financed by Greece and the European Union (European Social Fund- ESF) through the Operational Programme «Human Resources Development, Education and Lifelong Learning» in the context of the project "Strengthening Human Resources Research Potential via Doctorate Research" (MIS-5000432), implemented by the State Scholarships Foundation (IKY)» and supported by the A. G. Leventis Foundation scholarship. Support was provided also from the Stavros

Niarchos Foundation (SNF) in the form of a student scholarship. KT acknowledges funding from the European Research Council (ERC) under the European Unions Horizon 2020 research and innovation programme under grant agreement No. 771282 and support from the Foundation of Research and Technology - Hellas Synergy Grants Program through project POLAR, jointly implemented by the Institute of Astrophysics and the Institute of Computer Science.



**Data availability:** All data used for the specific study can be found in the NOA-ReACT Zenodo repository (https://zenodo.org/record/7233498#.Y5yUvXZBzIU) under CC-BY-SA (CC) rights. A detailed repository with general information about the instrument operation, data curation and supplementary material is publicly accessible in GitHub: https://github.com/NOA-ReACT/SolPol, also contains the present data processing algorithm version written in Python, under GPL-3.0 license.


**Conflicts of Interest:** At least one of the (co-)authors is a member of the editorial board of Atmospheric Measurement Techniques.

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



**(2) Linear Polarizer**

**(1) PEM head & Aperture**

**(4) Filter Wheel**

**(7) CCD camera**

**(3) Rotator**

**(5) Optics**

**(8) Astronomical mount**

**(6) Photodiode detector**

**(9) Peripherals**

**Figure 1: The SolPol direct sun solar polarimeter comprising of: (1) the Photoelastic Modulator Head (PEM) with the appropriate aperture, (2) the Linear Polarizer, (3) the assembly Pyxis Rotator, (4) the rotating Filter Wheel with the 550 nm centred narrowband filter, (5) the system Optics consisting of a Galilean telescope, (6) the 1-cm silicon Photodiode Detector. The instrument is mounted on (8), which is an EQ3 SynScan Astronomical mount and completed by (9) the essential Peripherals for the assembly operation, rotation and signal modulation. SolPol is operated in a 3-m astronomical dome installed at the PANGEA observatory in Antikythera, Greece.**



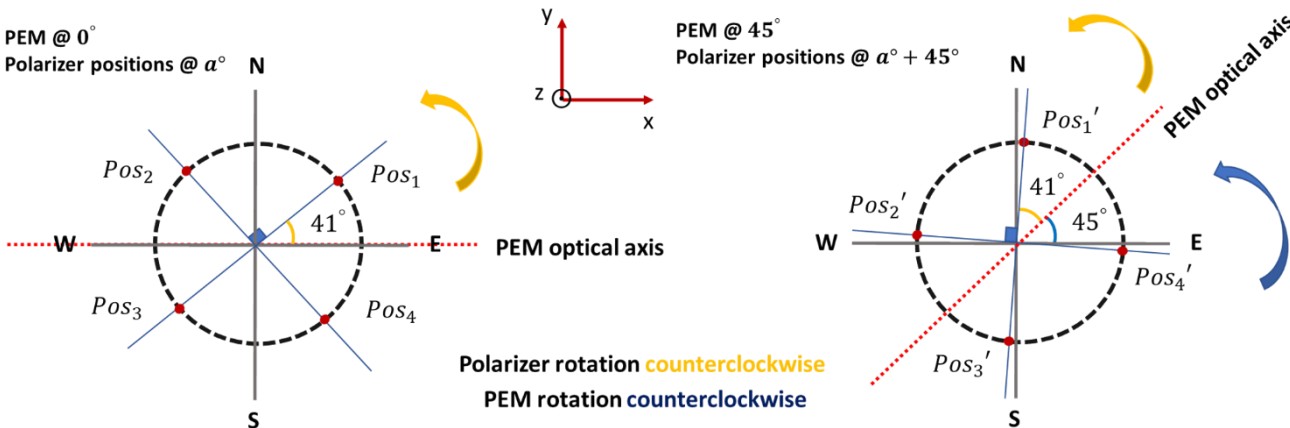

**Figure 2: SolPol assembly positioning as seen from the incoming sunlight reference frame. Left arrangement: The initial sequence begins with the PEM and linear polarizer at the rest position of 0°, then the polarizer rotates by 90° starting from 41° (Pos₁), to 131° (Pos₂), 221° (Pos₃) and 311° (Pos₄) counter-clockwise. A complete polarizer rotation with the PEM at 0° provides measurements of the U Stokes parameter. Right arrangement: assembly rotation by 45° and subsequent similar rotation of the polarizer by 90°**
**intervals from Pos₁' to Pos₄'. This configuration provides measurements of the Q Stokes parameter.**


**Table 1: Measurements of the Stokes parameters, from DC and AC measurements as a function of the modulation frequency, linear polarizer position angle and assembly rotation.**

| n | $I'$ | Assembly without rotation | Assembly rotated at 45° |
|---|---|---|---|
| 0 | DC | $\frac{1}{2}(I + Q\cos 2a + UJ_0(A)\sin 2a)$ | $\frac{1}{2}(I - QJ_0(A)\sin 2a + U\cos 2a)$ |
| 1 | 1ωt | $VJ_1(A)\ \sin\omega t\ \sin 2a$ | $VJ_1(A)\ \sin\omega t\ \sin 2a$ |
| 2 | 2ωt | $UJ_2(A)\cos 2\omega t\ \sin 2a$ | $-QJ_2(A)\cos 2\omega t\ \sin 2a$ |

**Figure 3: SolPol measurement on August 29th 2020, spanning from 06:00 to 16:00 UTC at the PANGEA observatory, Antikythera.**
**The day is labelled as "clean day" due to the very low aerosol content; zero percent cloud-cover and the acquired measurements**
**consist a "background reference measurement" for the instrument. In panels: (a) the Degree of Linear Polarization (DOLP) plotted**
**in parts per million (ppm) for four different test cases: calculated without subtracting the dark DC voltage for the specific day (black**
**dots), by subtracting the maximum value (purple dots), average (blue dots) and minimum (red dots) values of the dark DC voltage**
**output. The dash-dotted grey line signifies the instrument noise level at 50 ppm, (b) a zoomed representation of (a), (c) the total**
**incoming sunlight intensity, $v_{dc}$ in Volts as recorded by the lock-in amplifier. Different colours signify the corresponding**
**measurement subsets within the day and (d) the recorded wind speed in m/s for 30-minute averages on the specific day. During the**
**SolPol measurements generally low-winds were blowing over the station.**






**Figure 4: Complete SolPol linear polarization measurements acquired between May 2020 to June 2021 at the PANGEA observatory, Antikythera. Each daily observation consists of the normalized Q/I (black dots), U/I (purple dots) and DOLP (blue dots) in ppm with time, while labels D: stands for dust driven days with various loads, C: for clean days with low aerosol content, HC: for half-clean days where dust outbreaks occurred at some point within the day, and TD: is a dedicated test day with alternating iris sizes.**

**Mean AOD level 1.5 values are provided for each day by AERONET (https://aeronet.gsfc.nasa.gov/, last visited: 12/12/2022). DOLP values during clean (C) days determine the instrument noise level for the polarization measurements, while during dust events (D) the DOLP values peak early in the morning and in the afternoon, reaching polarization values of more than 200 ppms per case.**




**Figure 5: Top panel: Daily progression of linear polarization (in ppm) for the measurements on 29/08/2020, at the PANGEA observatory in Antikythera, expressed through the normalized Q/I (black dots), and U/I (purple dots) Stokes parameters and DOLP (blue dots). All values do not exceed the 50 ppm threshold regardless of the instrument viewing angle. Bottom panel: Time-height plot of the Volume Linear Depolarization Ratio ($\delta_v$) at 532 nm with the altitude, as retrieved from the Polly$^{XT}$ lidar at PANGEA for the same day. Low $\delta_v$ values denote that there is low concentration of depolarizing particles for the total duration of the SolPol measurement.**







**Figure 6: Top panel: Daily progression of linear polarization (in ppm) for the measurements on 02/09//2020, at the PANGEA observatory in Antikythera, expressed through the normalized Q/I (black dots), U/I (purple dots) Stokes parameters and DOLP (blue dots). The increase in linear polarization is exhibited before 08:00 and after 14:00 UTC when the instrument viewing angle increases significantly with respect to the zenith. Bottom panel: Time-height plot of the Volume Linear Depolarization Ratio ($\delta_v$) at**
**532 nm, as retrieved from the Polly$^{XT}$ lidar at PANGEA for the same day. High $\delta_v$ values (>15%) are indicative of dust particle presence and larger concentrations of dust are denoted with dark orange hues.**

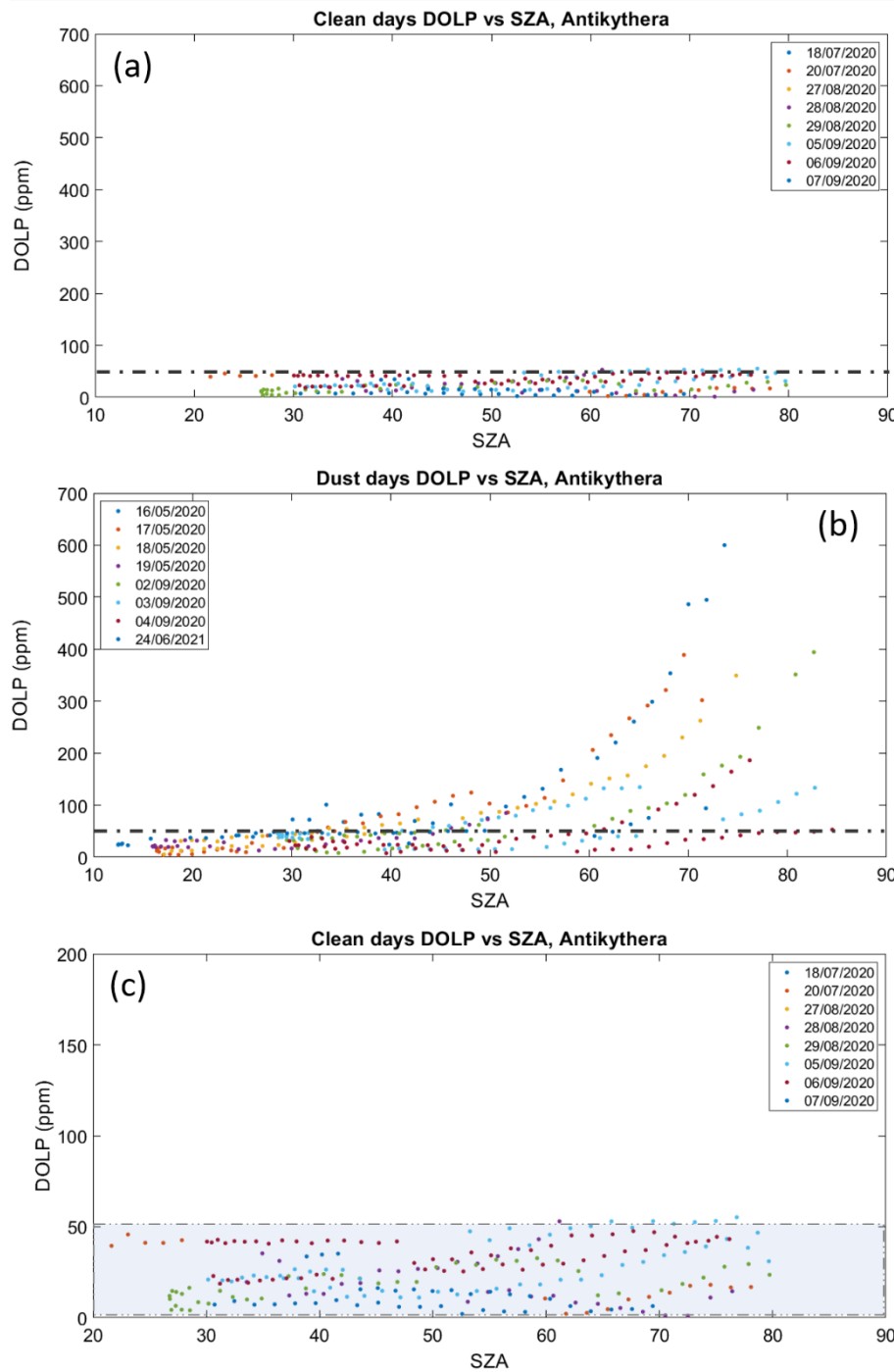

**Figure 7: Analysis of the DOLP retrievals with the solar zenith angle (SZA, in degrees) for the labelled: (a) clean days (C), (b) days with dust events (D), and (c) a zoomed representation of (a) with the instrument noise threshold (shaded) as deduced by the clean days. The excess in linear polarization (> 50 ppm) becomes apparent for large SZAs (> 50°) when dust particle concentrations persist within the day.**



**Figure 8: (a) DOLP as a function of the AOD with respect to the SZA, for the all the SolPol measurements in 2020 and 2021, at the PANGEA observatory. Star markers denote dust days (D), circle markers denote clean days (C) and the colour scale referes to different AOD ranges. For small AOD values (< 0.1), the linear polarization values are within the clean day threshold, while for AODs between 0.1-0.2, the DOLP values begin to exhibit the increasing trend with increasing SZAs, due to the potentially small concentration of preferentially oriented particles along the instrument line of sight. Significant linear polarization values are observed for all the dust days and the trend intensifies further for AODs towards the range between 0.4-0.6. (b) DOLP as a function of the airmass for the same AOD ranges. A linear correlation is observed between DOLP and the airmass for AOD values above 0.1, for the dust cases. For larger AODs, the DOLP values surpass the noise threshold for even smaller airmasses.**

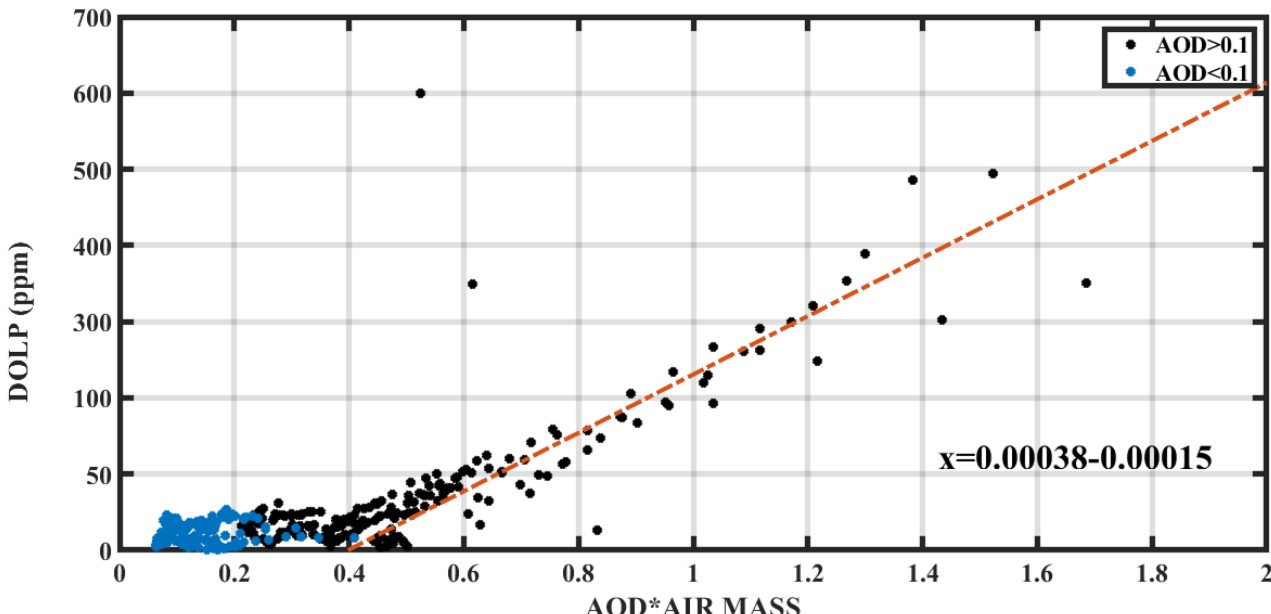

**Figure 9: Scatterplot of DOLP as a function of the slant AOD, for the SolPol measurements in Antikythera. The linearity observed on AODs over 0.1 corresponds to DOLP signatures from dust days only.**



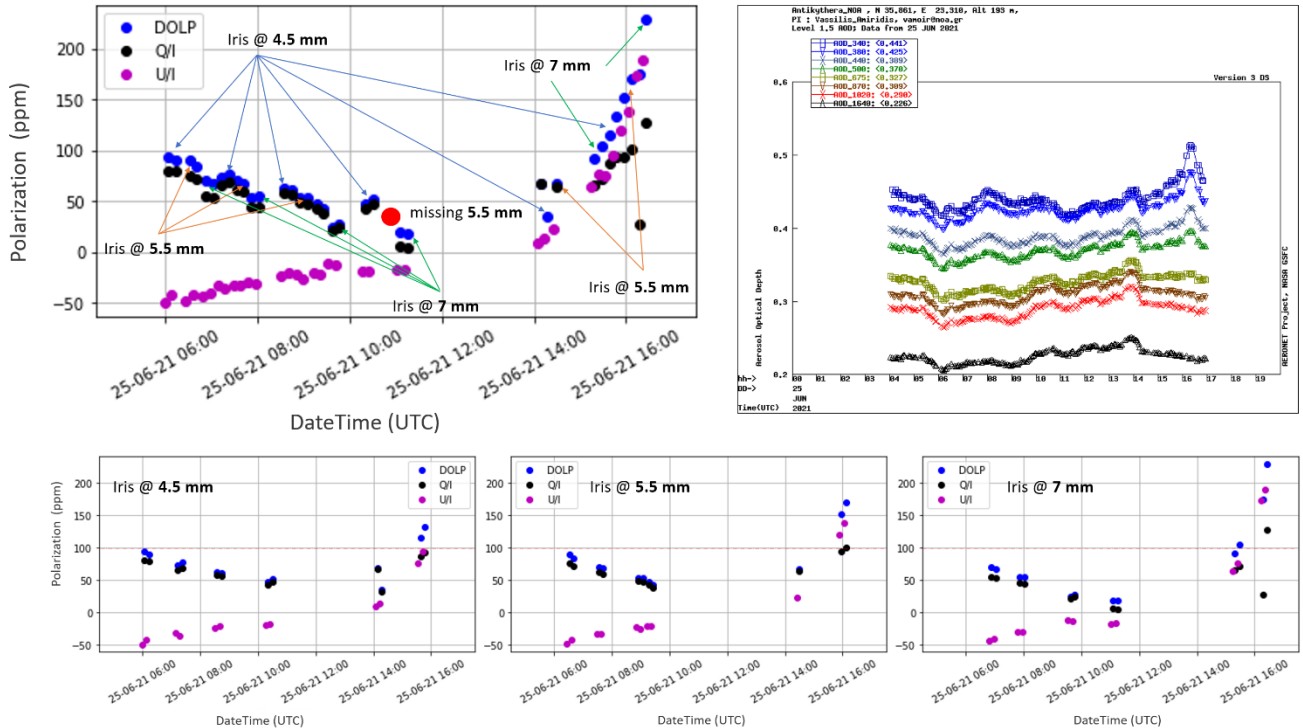

**Figure 10: Alternating iris size tests for the quantification of diffuse light contribution to the linear polarization observations under the 25/06/2021 dust layer, with an AOD of 0.370 at 500 nm. Top pannels present the normalized Q and U Stokes parameters, DOLP and the different colour arrows denote the respective iris size for the each measurement set, along with the AOD progression witthin the day from AERONET.**



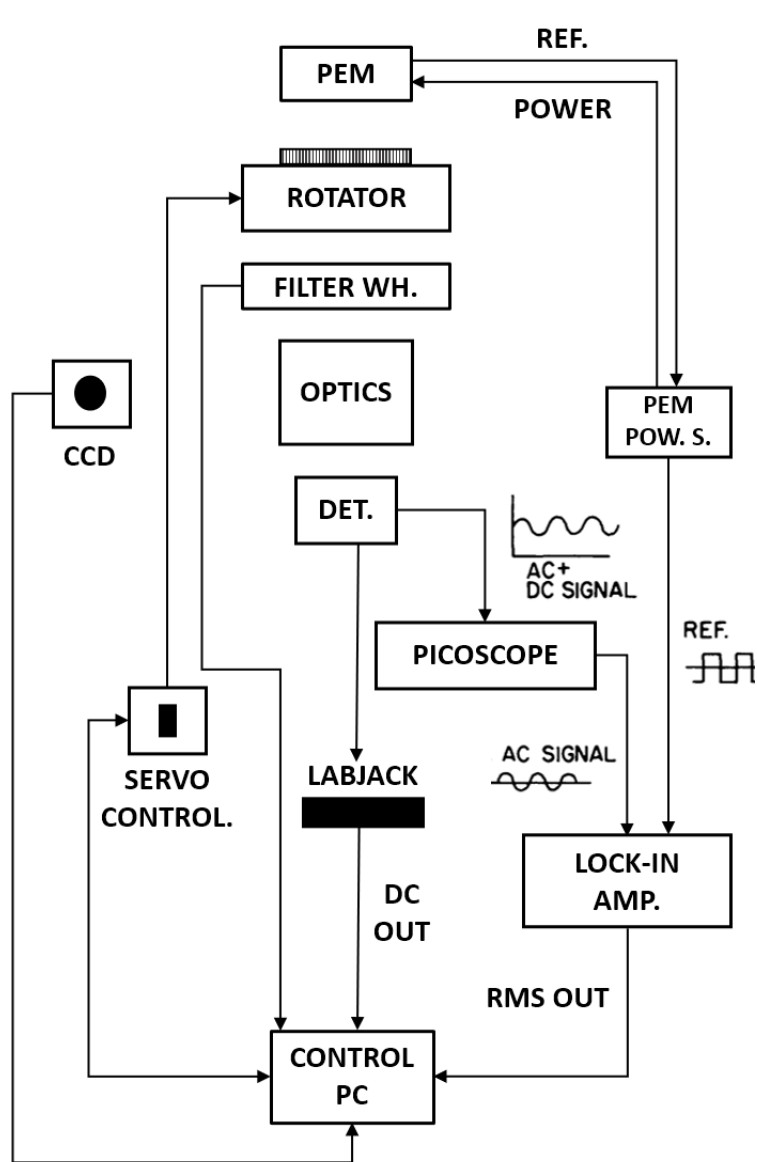

Figure A1: SolPol electronics and data acquisition diagram.



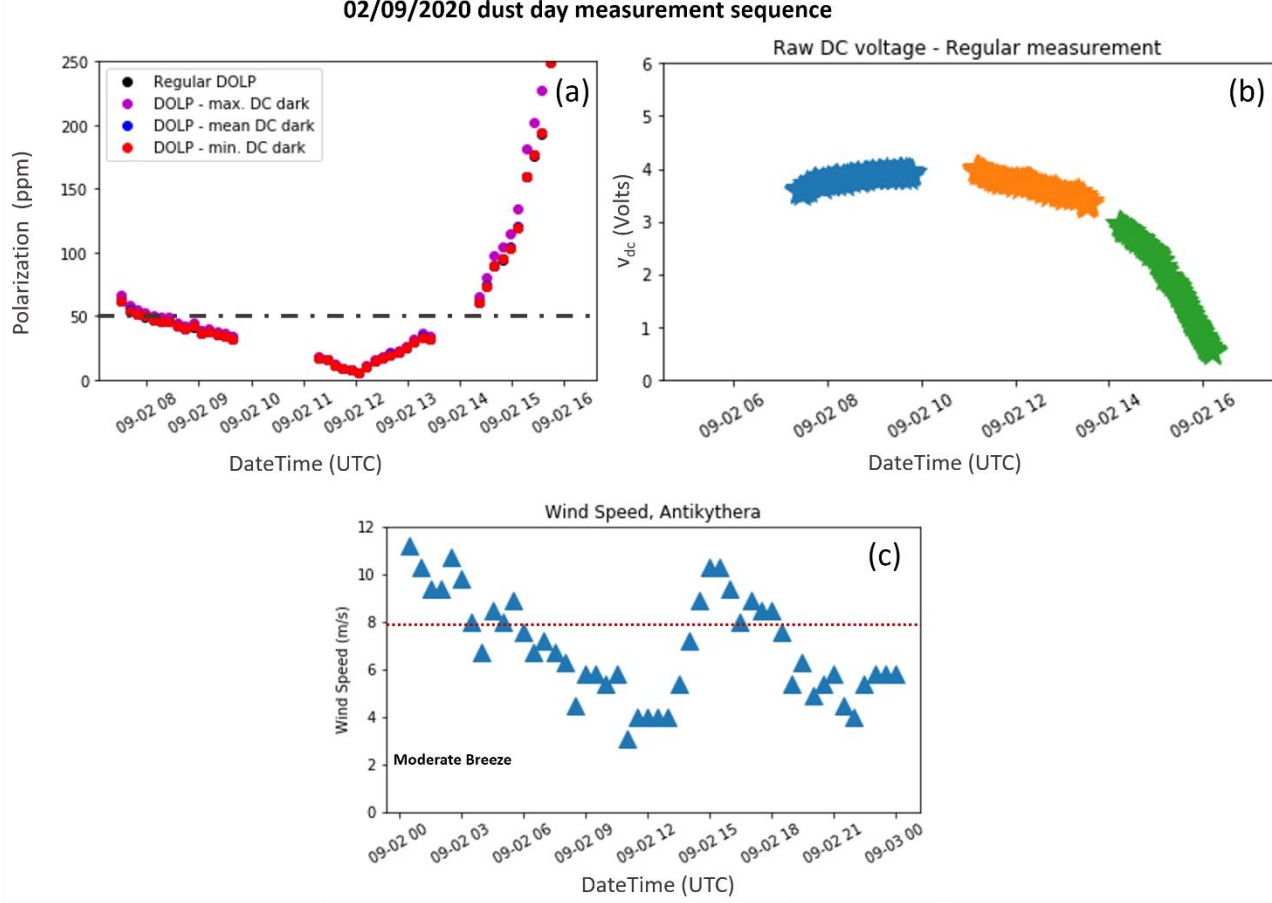

810

**Figure A2: SolPol daily measurement on 02/09/2020, spanning from 07:00 to 16:00 UTC at the PANGEA observatory, Antikythera. The specific day is affected by a persistent dust outbreak above the station, has zero percent cloud-cover and consists a typical dust-driven measurement for the instrument. In panels: (a) the DOLP is plotted in parts per million in polarization for four test cases: calculated without subtracting the dark DC voltage for the specific day (black dots), by subtracting the maximum value (purple dots), average (blue dots) and minimum (red dots) values of the dark DC voltage output. The dash-dotted grey line signifies the instrument noise level at 50 ppm, (b) the total incoming sunlight intensity, $v_{dc}$ in Volts as recorded by the lock-in amplifier. Different colours signify the corresponding measurement subsets within the day. (c) The recorded wind speed in m/s for 30-minute averages for the specific day. During the SolPol measurement moderate winds were blowing over the station.**

**Table A1: Measured minimum/maximum dark DC signal intensity and the calculated average value for the selected days of bias**
820 **correction demonstration.**

| Dark DC (Volts) | 29/08/2020 (C) | 02/09/2020 (D) |
|---|---|---|
| $I_{min,dark}$ | 0.03095 | 0.00408 |
| $I_{mean,dark}$ | 0.04778 | 0.01148 |
| $I_{max,dark}$ | 0.16665 | 0.24016 |





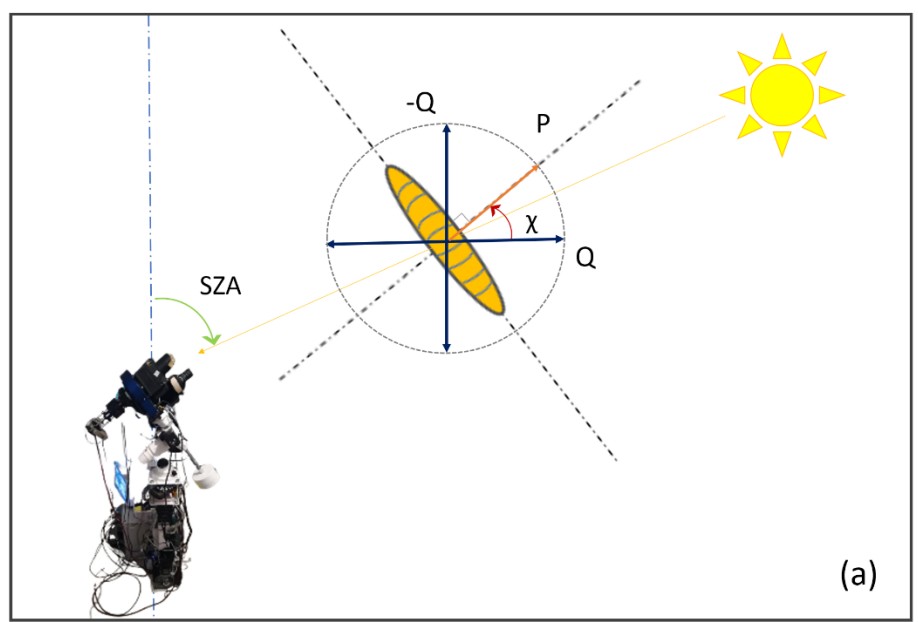

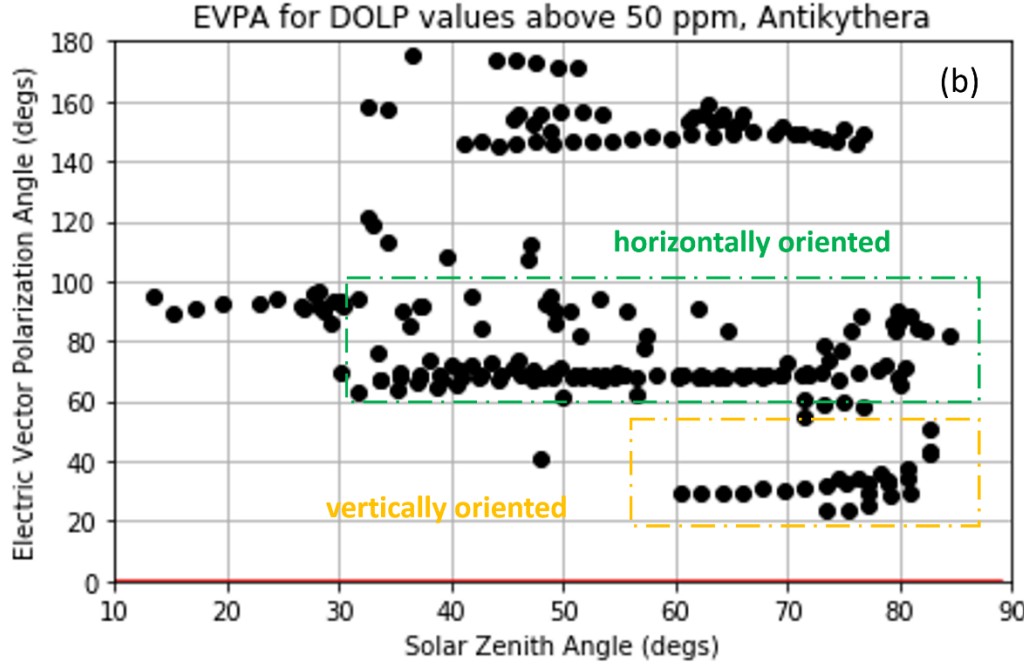

**Figure A3: (a)** SolPol viewing geometry for an elongated dust particle with respect to the solar zenith angle (SZA), **(b)** Electric Vector Polarization Angle (EVPA) in degrees with respect the SZA for all SolPol measurement days in 2020 and 2021. EVPA angles correspond to DOLP values > 50 ppm. There is a clear distribution of angles in two distinct clusters around 20° and 70° that correspond to preferentially vertically (yellow tab) and horizontally (green tab).