# Peer review of "Linear Polarization Signatures of Atmospheric Dust with the SolPol direct sun polarimeter"

_Atmospheric Measurement Techniques, 2023_

## Author Comment (AC1)

**RESPONSE TO REFEREE#1 COMMENTS AND PEER-REVIEW REPORT**

Manuscript Title: **"Observations of Dust Particle Orientation with the SolPol direct sun polarimeter"**

Authors (as declared in the submitted manuscript):

**Vasiliki Daskalopoulou, Panagiotis I. Raptis, Alexandra Tsekeri, Vassilis Amiridis, Stelios Kazadzis, Zbigniew Ulanowski, Vassilis Charmandaris, Konstantinos Tassis and William Martin**

The following affiliation has been added to Prof. Vassilis Charmandaris as:

"[8]**European University Cyprus, Diogenes St., Engomi, Nicosia 1516, Cyprus**"

Dear respected Editor/Reviewers,

The authors highly appreciate the comprehensive feedback throughout the review process and kindly reply to the reviewer comments, as follows:

**REFEREE#1 COMMENTS:**

"*The paper by Daskalopoulou et al. introduced the very interesting and important observation of polarization in the direct sun transmission measurement which can prove the preferred orientation of dust particles in the atmosphere. The paper provides a clear description on the experimental setup as well as data acquisition and processing. I read this work with much interests and have a few comments for the authors to consider or confirm:*

*1. I'm not quite sure about the proof of little contribution of diffuse light scattering to the observed DOLP in the transmitted measurements. DOLP is the ratio of polarized intensity against intensity. So even if the aperture size increases, both intensity and polarized intensity could increase so that the ratio remain not much impacted. Could the author elaborate more on this?*"

**Author's Response:** The authors are grateful to the respected reviewer for the comments about the paper readability in its preprint state. Concerning the first comment on diffuse light contribution to our measurements, we would like to initially state that the testing that we could perform on ambient conditions with SolPol was limited only to empirical measurements by increase/decrease of the incoming diffuse light flux when selecting different aperture sizes. The test procedure comprised of alternating iris sizes from small (at 4.5 mm), to regular (5.5 mm, i.e., the one used for regular SolPol measurements) to large (at 7 mm) through consecutive measuring intervals so as to ensure that polarization would not change significantly with the instrument viewing angle. The rationale was that with larger iris, more light is measured by the instrument and by increasing the diffuse light in the solid angle. Hence, if there is

contribution of the diffuse light scattering to the measured linear polarization, then the latter should also increase with a larger iris. For a simple adjustable aperture stop as the one used in the current setup, when increasing its diameter, we increase the effective area and therefore the incoming total solar flux. By examining the following figure (Fig. 1, as opposed to Figure 10 - Section 5.3 in the manuscript), we focus on the early morning to noon measurements, when AOD is relatively constant and compare the first 7 mm measurement to the following (second individual) 4.5 mm measurement (both in red squares). As we can see, DOLP increases from larger iris to the smaller iris while we would expect the opposite to happen in the case of strong contribution from the diffuse light to the linearly polarized light or at least that the DOLP ratio would remain constant provided that the decrease in intensity when downsizing is proportionate to the decrease in polarized intensity. Although there is a detailed description in the SolPol manual pg. 13, we have added the following sentence in the manuscript and hope this clarifies the discussion on Section 5.3.

Lines 466 - 467: "**We focus on the early morning to noon measurements, when AOD is relatively constant and compare the first 7 mm measurement to the following (second individual) 4.5 mm measurement.** As seen in **Error! Reference source not found.**,..."

Lines 471 - 472 are rephrased as follows: "**DOLP increases from larger iris to the smaller iris while we would expect the opposite to happen in the case of strong contribution from the diffuse light to the linearly polarized light received by the detector**. This proves that by increasing the…"

[Figure]

**Figure 1:** Alternating iris size tests for the quantification of diffuse light contribution to the linear polarization observations under the 25/06/2021 dust layer, with an AOD of 0.370 at 500 nm. Top panels present the normalized Q and U Stokes parameters, DOLP and the different colour arrows denote the respective iris size for each measurement set, along with the AOD progression within the day from AERONET. Bottom panels present the individual measurement sequences for each iris size, while red squares denote the succession from larger iris to the smaller one.

*"2. The Rayleigh optical depth is ~0.14 for 500nm. Under clear sky conditions, the diffuse Rayleigh should contribute some signals to DOLP. This will bring minor but potential contamination via multiple scattering which has certain dependence on solar angle. From the Fig. 7a, however, it looks the contribution from Rayleigh to be very small (<50ppm) and there is little dependence on solar angle. Could the authors confirm this is the case? Is it because the direct transmission is very strong so that DOLP is further diluted? To have a better view, the authors may try separating the direct direction of sunlight and scattering contribution given that the solar irradiance is known at 500 nm and the optical depth for both Rayleigh and aerosol (from AERONET) is known. Then the DOLP contributed by diffuse scattering could be better observed."*

**Author's Response:** As is presented in Figure 7a and the zoomed representation in (c), the measured Rayleigh scattering contribution in the forward direction and under clear conditions (no dust particle presence indicated by the lidar retrievals) to DOLP is less than 50 ppms, when we use the default 5.5 mm aperture size. It also appears to have no dependence on the solar zenith angle for the narrowband filter at 550 nm. Compared to the DOLP values we are observing when dust particles are present, the latter are almost one order of magnitude larger than the clean day cases. If we increase the iris diameter, Rayleigh contributions to the polarization fractions at 550 nm can increase by a factor of two, which is consistent with predicted values for Rayleigh scattering at low altitudes (e.g., Mishchenko et al., 1994 and references therein). Since all measurements are direct sun and we ensure a stable tracking of the sun disk without misalignments, incoming diffuse light within our FOV from scattering angles close to 0° (forward scattered light) will not contribute to DOLP and, in fact, linear polarization will decrease with increasing AOD due to further loss of polarization from multiple scattering (see adapted Figure 2 from Hansen and Travis, 1974). Furthermore, these changes due to multiple Rayleigh scattering are found in diffuse light, and for direct solar irradiance only the optical depth affects the beam. Since we are confident that the residual contribution of diffuse light to the measured signal (as explained in the previous question), is very small, the zenith angle independence of DOLP at 50 ppm is explained.

We thank the reviewer for their suggestion concerning the DOLP contribution distinction by utilizing the AERONET data. In order to make these estimations an absolute calibration of the irradiance measurements would be required, in order to separate the two components from theoretical calculations. The setup of this study had a different aim and the constant changes of integration times and apertures would require independent calibrations for each set. We plan for future experiments to include these calibrations in order to make possible this kind of analysis.

[Figure]

**Figure 2:** Linear polarization, -100 Q/I, as a function of the phase angle for different aerosol optical depths (figure adapted from Hansen and Travis, 1974).

**References:**

Hansen, J.E., and L.D. Travis, 1974: Light scattering in planetary atmospheres. Space Sci. Rev., 16, 527-610, doi:10.1007/BF00168069.

Mishchenko, M. I., Lacis, A. A. and Travis, L. D.: Errors induced by the neglect of polarization in radiance calculations for rayleigh-scattering atmospheres, J. Quant. Spectrosc. Radiat. Transf., 51(3), 491–510, doi:10.1016/0022-4073(94)90149-X, 1994.

*"3. I wonder whether one likely cause of very small DOLP (<700ppm) to be the short wavelength (500nm) the authors are experimenting whereas the dust particles are coarse (>5 microns). How difficult will it be to switch to a larger wavelength (e.g. 1020nm or even infrared) and repeat the measurement. This way the authors may obtain higher sensitivity of DOLP to the orientation of large particle size."*

**Author's Response:** We thank the reviewer for their insightful comment on considering switching to longer wavelengths, so as to maximize the potential polarization signature for dust. Mineral dust is often described as "white" or "gray" in terms of its wavelength response. As an example, Bailey et al. (2008) have discussed the only logical question if larger dust particles give polarization signatures in the same manner as small dust particle, which are mostly responsible for interstellar polarization. As seen in Figure 3 (adapted Figure 4 from the respective paper of Bailey et

al., 2008), where the T-matrix calculated ratio of the total extinction coefficient to the dichroism, $K_{12} / K_{11}$, is presented as a function of the size parameter χ, for larger particles we are in the plateau regime where the mean polarization is essentially the same as that produced by smaller particles. Since larger particles are the ones that will eventually become preferentially oriented by overcoming randomization due to Brownian motion (e.g., Mallios et al., 2021; Ulanowski et al., 2007), we do not expect to observe a significant contribution to the phenomena by switching to longer wavelengths.

[Figure]

**Figure 3:** Ratio of extinction matrix elements $K_{12}/K_{11}$ for vertically oriented prolate spheroidal particles of axis ratio 1.8 at a zenith distance of 60∘ as a function of size parameter ($x = 2\pi r/\lambda$). The lower panel shows the full range of variation including the peak corresponding to the small particle regime that causes interstellar polarization. The upper panel is on an expanded scale showing the oscillatory nature of the polarization at larger sizes, but also that the mean level is positive (horizontal polarization). The particle size at a wavelength of 0.8μm is shown on the top scale.

In terms of technical limitations, SolPol is an experimental instrument and its complete characterization has already been painstakingly long. Some of the tasks to be tackled would be the need to consider new communication protocols between instrument peripherals in order to operate at an additional wavelength and, most importantly, integrate a new photodiode detector operating in the NIR with the capability of stably detecting small signals. The responsivity and quantum efficiency of photodiodes decrease with increasing wavelength in the NIR range, meaning that the sensitivity decreases, resulting in lower signal-to-noise ratios and reduced detection efficiency. The limit for reasonable signals from the specific photodiode is about 750-800 nm with a good SNR. When approaching wavelengths beyond 1000 nm it is a very different experiment than what we have setup so far, with encroaching telluric absorption lines beginning to come into play. Going towards the lower wavelengths' direction, i.e., 450 nm might also be interesting in comparison to our previous 550 nm. Instrument filters on the 450 and 750 nm central wavelengths have been set up in the past for lab measurements and Rayleigh contribution field measurements in Hatfield, but definitely the

bandwidth of these filters will make a difference in ambient field conditions in Antikythera. Thermal noise is also a significant source of error to be considered in these wavelengths, thus the trade-off led initially to the use of the instrumental configuration as is.

Nonetheless, we again thank the reviewer for the kind suggestion and we will revisit the SolPol assembly configuration in the NIR as a potential upgrade to be completed in the near future and provide complementary data on dust orientation from various wavelengths.

**References:**

Bailey, J., Ulanowski, Z., Lucas, P. W., Hough, J. H., Hirst, E. and Tamura, M.: The effect of airborne dust on astronomical polarization measurements, Mon. Not. R. Astron. Soc., 386(2), 1016–1022, doi:10.1111/j.1365-2966.2008.13088.x, 2008.

Mallios, S. A., Daskalopoulou, V. and Amiridis, V.: Orientation of non spherical prolate dust particles moving vertically in the Earth's atmosphere, J. Aerosol Sci., 151(August 2020), 105657, doi:10.1016/j.jaerosci.2020.105657, 2021.

Ulanowski, Z., Bailey, J., Lucas, P. W., Hough, J. H. and Hirst, E.: Alignment of atmospheric mineral dust due to electric field, Atmos. Chem. Phys., 7(24), 6161–6173, doi:10.5194/acp-7-6161-2007, 2007.

---

## Author Comment (AC2)

**RESPONSE TO REFEREE#2 COMMENTS AND PEER-REVIEW REPORT**

Manuscript Title: **"Observations of Dust Particle Orientation with the SolPol direct sun polarimeter"**

revised per **reviewer#2** comments as:

**"Linear Polarization Signatures of Atmospheric Dust with the SolPol direct sun polarimeter"**

Authors (as declared in the submitted manuscript):

**Vasiliki Daskalopoulou, Panagiotis I. Raptis, Alexandra Tsekeri, Vassilis Amiridis, Stelios Kazadzis, Zbigniew Ulanowski, Vassilis Charmandaris, Konstantinos Tassis and William Martin**

The following affiliation has been added to Prof. Vassilis Charmandaris as:

"[8]**European University Cyprus, Diogenes St., Engomi, Nicosia 1516, Cyprus**"

Dear respected Editor/Reviewers,

The authors highly appreciate the comprehensive feedback throughout the review process and kindly reply to the reviewer comments, as follows:

**REFEREE#2 COMMENTS:**

**Major comments**

*"This paper reports direct sun observations of the three Stokes vector components ...*

*1.        There is, however, an important point and it is the fact this paper is titled as an investigation to suggest particle orientation whereas the evidence shown is at best consistent with the fact but by no means the only possibility. Alternate possibilities are not discussed and there are several statements in the text seem too biased to confirm that particle orientation is occurring. As it is now the text reads assuming that particles are oriented, and the discussions are biased towards this end. With this regard, I advise to change the title of the paper without mentioning particle orientation but rather emphasize the novelty of the instrument for atmospheric applications and reported observed DLP for atmospheric dust. I consider this aspect as major point as I do not consider it should be published with such title. I believe this change and some changes in the tone of the text should be relatively simple."*

**Author's Response:** The authors are deeply grateful to the respected reviewer for the effort given and the thorough analysis on the paper key points, as the feedback has been invaluable in refining the content and strengthening the overall quality of the paper. Specifically addressing the crucial point on the paper title, we value the insightful comment which indeed would describe the context in a broader manner and will not predispose the reader towards a single strong elucidation of the reported DOLP signatures. For that matter, we have modified the manuscript title as suggested further in the review, to:

"**Linear Polarization Signatures of Atmospheric Dust with the SolPol direct sun polarimeter**"

**Minor/Editorial comments**

2.      *"Abstract and Lines 31-32: The concept of orientation is only mentioned in the introduction and in the last line in a suggestive sentence (as opposed to assertive sentence) ... with a novel sunphotometer.")"*

**Author's Response:** addressed in previous comment no1.

3.      *"Line 63: change to "The latter was addressed ..."*

**Author's Response:** Thank you, we corrected it accordingly.
Corrected "*The latter is addressed*..." to "***The latter was addressed***..." as indicated.

4.      *Lines 49 to 65: The contents and source of information of this paragraph is adequate. However… So, it would be very useful to acknowledge this fact with some text tweaking and text additions (probably much less text than what I just wrote here)."*

**Author's Response:** First of, we would like to thank the reviewer for rigorously reading the introduction and providing helpful insight to the paragraph readability and content bridging. As we state in the manuscript (**Lines 82-84**), this work strives to reproduce the observations summed in the two referenced studies for atmospheric dust orientation from Ulanowski et al., (2007) and Bailey et al., (2008), respectively, with the major differentiation and challenge of using direct daylight measurements. We generally agree with the reviewer that the alignment mechanisms of atmospheric dust are not as straightforward as in the case of intergalactic dust, but they are nonetheless theoretically predicted in orientation models. As an example, Mallios et al., (2021) rigorously examine the alignment of charged/uncharged dust particles under the effect of the large scale electric field that is present in the Earth's atmosphere due to the potential difference between the lower part of the ionosphere and the Earth's surface. This field creates an electrical torque that acts upon the particles and can influence their transport dynamics, therefore their

orientation which is depended on the particle size (and shape). Other mechanisms that affect particle orientation are the bombardment of gas particles that tend to randomize the orientation (rotational Brownian motion) and the aerodynamic torque that emerges due to the fact that the drag force acts on the center of pressure of the particle which is different than the center of gravity, and tends to orient the particles horizontally. The electrical torque, on the other hand, tends to orient the particles vertically, according to the study, depending on the particle's charge and size and for electric field strengths larger than 10 kV/m where vertical orientation is possible (Mallios et al., 2021). These required fields are orders of magnitude larger than the typical electric field value or the reported values on elevated layers (e.g., Daskalopoulou et al., 2021b), but were previously measured in dust storms close to the dust source (review by Riousset et al., 2020 and references therein). The aforementioned information is stated in the introduction (**Lines 66-81**) in order to provide the theoretical plausibility of atmospheric dust orientation (considering the much smaller size of dust than ice crystals, for example, that exhibit clear orientation signatures), through electrostatic means and to state that also other effects on particle motion, such as turbulence (e.g., Klett J. D., 1995), may result in preferential orientation but remain to be examined both by modelling and accompanied observations.

[Figure]

**Figure 1:** Ratio of extinction matrix elements $K_{12}/K_{11}$ for vertically oriented prolate spheroidal particles of axis ratio 1.8 at a zenith distance of 60° as a function of size parameter ($x = 2\pi r/\lambda$). The lower panel shows the full range of variation including the peak corresponding to the small particle regime that causes interstellar polarization. The upper panel is on an expanded scale showing the oscillatory nature of the polarization at larger sizes, but also that the mean level is positive (horizontal polarization). The particle size at a wavelength of 0.8μm is shown on the top scale.

In terms of the differentiation of aligned particle optical properties in contrast to non-aligned particles, we rely on the previous modelling work of Bailey et al., (2008) who investigated the expected polarization for spheroidal particles via T-matrix calculations of the forward scattering (extinction) matrix. A range of particle sizes, aspect ratios and

zenith distances were used for a particular particle composition, so although the study focuses on a longer wavelength, for size parameters that correspond to particles larger than 3µm for a 550 nm wavelength, there is an expected excess in linear polarization for vertically or horizontally aligned particles (see adapted Figure 1 from Bailey et al., 2008).

By carefully considering the highlighted distinction between the intergalactic dust and its atmospheric counterpart, we rephrase **Lines 56** to **66** and add the following sentences:

*"Dichroism measurements provide information on the magnetic field orientation, which is the dominant alignment mechanism **for these sub-micron particles** (Andersson et al., 2015; Dasgupta Ajou K., 1983; Kolokolova and Nagdimunov, 2014; Lazarian, 2007; Siebenmorgen, 2014; Skalidis and Tassis, 2020). **Concerning the much larger atmospheric dust particles, the geomagnetic field is considered a weak alignment mechanism, since multiple processes such as the bombardment by gas particles, the imposed aerodynamic and electrical torques (Mallios et al., 2021; Ulanowski et al., 2007a and references therein), but also turbulence (e.g. Klett J.D., 1995) compete (or counter-balance) for the most dominant atmospheric particle alignment mechanism. Based on a similar optical approach, though**, atmospheric dust may provide distinct linear polarization (LP) signatures, as vertically oriented particles can lead to dichroic extinction of the transmitted sunlight. This was indicated for starlight observations during nighttime, which showed predominantly horizontally polarized light during a Saharan dust episode in La Palma (Bailey et al., 2008; Ulanowski et al., 2007b). **Also in the same study, modelling of the forward scattering matrix through T-matrix calculations was employed and showed that excess polarization for spheroidal particles of a specific composition and orientation is to be expected for particle sizes larger than 3 µm.** However, the discussed measurements refer to column-integrated values that are not capable of resolving the vertical distribution of the phenomenon throughout the dust layer. The latter was addressed by a novel polarization lidar for detecting dust orientation, nicknamed WALL-E, which is expected to provide valuable information for monitoring the phenomenon of dust polarization in the Earth's atmosphere (Tsekeri et al., 2021).
**In order to provide the theoretical feasibility of orientation**, a potential mechanism that is capable…"

Regarding the reviewer's suggestion/inquiry on "what is the connection and consequence of not having particle random orientation and the observed optical properties?", the origin of polarization of direct sunlight is already explained in the Introduction (Lines 49-54) and further discussion can be found in the references cited (Ulanowski et al. 2007, Bailey et al. 2008). See also response to Comment 5.

**References:**

Bailey, J., Ulanowski, Z., Lucas, P. W., Hough, J. H., Hirst, E. and Tamura, M.: The effect of airborne dust on astronomical polarization measurements, Mon. Not. R. Astron. Soc., 386(2), 1016–1022, doi:10.1111/j.1365-2966.2008.13088.x, 2008.

Klett, J. D., 1995: Orientation Model for Particles in Turbulence. J. Atmos. Sci., 52, 2276–2285, https://doi.org/10.1175/1520-0469(1995)052<2276:OMFPIT>2.0.CO;2.

Riousset, J. A., Nag, A., & Palotai, C. (2020). Scaling of conventional breakdown threshold: Impact for predictions of lightning and TLEs on Earth, Venus, and Mars. Icarus, 338, Article 113506. http://dx.doi.org/10.1016/j.icarus.2019.113506.

Ulanowski, Z., Bailey, J., Lucas, P. W., Hough, J. H. and Hirst, E.: Alignment of atmospheric mineral dust due to electric field, Atmos. Chem. Phys., 7(24), 6161–6173, doi:10.5194/acp-7-6161-2007, 2007.

*5.        "Also, in my search of bibliography ... Chapters 5-7 are also very relevant: Coulson, Kinsell L. "Polarization and Intensity of Light in the Atmosphere." (1989). A. Deepak Pub., Hampton, Va.,"*

**Author's Response:** The reviewer correctly states that not many direct sun (i.e., dichroic extinction) polarization measurements do exist, but then mentions some recent papers describing all-sky measurements. Indeed, in contrast to the direct measurements, there is a huge body of past work focused on all-sky (diffuse) polarization. Unfortunately, such measurements have limited relevance to the present study, which is focused on particle alignment. The reason is that the principal cause of dichroic extinction is particle alignment, while the interpretation of diffuse polarization measurements in terms of particle alignment is highly ambiguous, because multiple causes contribute to sky polarization. Hence it is much harder to derive information on particle alignment from sky, as opposed to direct sun, polarization measurements. We thank the reviewer, though, for providing references to bibliography, based on all sky polarization measurements, and on some classic textbook information that will strengthen our knowledge on the subject. We were not aware of the very recently published papers by Li S. et al., (2023), Pan P. et al., (2023) and Guan L. et al., (2018) and we have incorporated them in the main text with the prospect of potentially being used as background information on future studies. Therefore, a paragraph is added as follows:

**Line 54 to 64:** "Interpreting linear polarization measurements in the direct direction (i.e., measurements taken when the observer looks directly towards the Sun) can be challenging due to the overwhelming intensity of sunlight and secondary sources of linear polarization in the observational line-of-sight. All-sky polarization patterns can serve as a background reference tool to aid in the interpretation of direct polarization measurements, as the direct direction is a subset of the former (Guan L. et al., 2018; Pan et al., 2023). Recent studies that compare radiative transfer simulations for aerosol media of well-defined optical properties with all-sky observations, show that the degree of linear polarization (DOLP) is impacted, and in fact decreases, by the increasing optical depth of desert dust particles (assumed spherical) even in the forward scattered direction (Li S. et al., 2023). However, the interpretation of diffuse polarization measurements in terms of particle alignment is highly ambiguous, because multiple causes contribute to

sky polarization. Hence it is much harder to derive information on particle alignment from sky, as opposed to direct sun, polarization measurements."

*6.        "Line 294 : Gassó et al, reference is wrongly listed. Check the proper reference."*

**Author's Response:** There was an unfortunate reference mix-up here, we have corrected it accordingly in the manuscript, thank you. The proper citation would be:

Gassó, S. and Knobelspiesse, K. D.: Circular polarization in atmospheric aerosols, Atmos. Chem. Phys., 22, 13581–13605, https://doi.org/10.5194/acp-22-13581-2022, 2022

*7.        "Line 316 : note the Kemp et al, paper also reports a non-zero V component in observed solar flx (of the order ~10-6)"*

**Author's Response:** Indeed, Kemp et al. (1987) report CP values of the order of ~ 3 x $10^{-6}$ in 550 nm, for the case of targeting specific cardinal co-ordinates in the Sun's disk (Fig. 2 in paper), with half the aperture angular diameter used in SolPol. These measurements could potentially be enhanced by regions of intense magnetic activity and strong area-dependent circular polarization near the solar limb. In contrast, in the whole Sun disk measurements (Fig. 3 in the same paper) CP values drop to the order of ~ $10^{-7}$ due to mainly the use of a larger aperture which would increase the observed effective diameter and subsequently, if the increase in circularly polarized flux (*V*) is not linear with the increase in total solar flux (*I*), the CP will decrease (V in Kemp's paper denotes the normalized *V/I* quantity).
The latter observations are consistent with what we observe with SolPol for the total solar disk in the forward direction, which is not negligible although near the limits of the instrument sensitivity. In order to extract the proper information for aligned atmospheric particles from CP (please refer to **Lines 293-300** in the preprint manuscript), we should focus on further processing the existing dataset and we tend to in future research, so we thank the reviewer for the remark.

**References:**

Kemp, J. C., Henson, G. D., Steiner, C. T. and Powell, E. R.: The optical polarization of the Sun measured at a sensitivity of parts in ten million, Nature, 326(6110), 270–273, doi:10.1038/326270a0, 1987.

*8.        "Line 365-366: Not a clear sentence, I do not understand what you are traying to say. Please rewrite/clarify as needed."*

**Author's Response:** We thank the reviewer for the comment and we will try to clarify. The sentence subtly refers to the particle viewing geometry from the instrument's reference frame and how the angle under which we perceive particle orientation changes with respect to the increasing solar zenith angle. From our understanding, we have attempted a schematic that depicts the viewing angles in SolPol's line-of-sight for a small collection of dt dust particles that are either i. horizontally oriented with respect to the ground and ii. vertically oriented with respect to the ground (Figure 2). As seen in the following figure, in both orientation cases (either horizontal or vertical) the instrument does not detect preferential alignment directly in the zenith, where SZA equals to zero, as specifically in the horizontal orientation case the azimuthal distribution of the grains is uniform and cannot be ignored. While, in the vertically aligned collection particles gradually appear to become "more" vertically aligned as we progress to larger SZAs. Hence, the sentence refers to either that differentiation in viewing angles and that as we progress to larger SZAs we would expect linear dichroism to be more prominent or to the increase of oriented particles as the instrument targets larger airmasses with the increasing zenith angle.

[Figure]

**Figure 2:** Four representative horizontally (top panel) and vertically (bottom panel) aligned particles for three distinct zenith angles, in the instrument's frame of reference.

Therefore, we have rephrased the sentence as follows:

**Lines 365 - 368:** the sentence "As the SZA increases, particles that are preferentially aligned with respect to the vertical axis present themselves in the frame of reference of the instrument as being more strongly aligned. Moreover, the airmass increases, hence the slant optical thickness and therefore dichroic extinction (which is an extensive scattering property – it grows with the number of scattering particles) also increase. Consequently, it is expected that at near-zenith angles, particles that are aligned with their long axis vertically will not influence the polarization and result in near zero values. Conversely, strong polarization is expected for large SZA, similarly to what was previously reported (Bailey et al., 2008)."

9.       *"Lines 335-337: "This is our first consistent indication ... These two facts warrant additional analyses as MS and molecular polarization can play an unknown role and are a consideration that probably were not necessarily in the cited studies."*

**Author's Response:** We refer the reviewer to the authors' answer on the first reviewer's comment no2. i.e. *"2. The Rayleigh optical depth is ~0.14 for 500nm ... diffuse scattering could be better observed"* and the Lines 446-451 of the submitted manuscript where we state that more tests are needed in order to further intensify our arguments. Taking into account the reviewer's kind suggestions, we change the tone of the following sentences:

**Lines 335 - 337** moved to **Line 341** and rephrased as: "This could potentially indicate that preferentially - vertically or horizontally - aligned dust particles might be present in the observed layers and cause the strong dichroic extinction of sunlight transmitted through the layer."

**Lines 379 - 381**: "The segregation between reference days and dust driven days, along with the distinct behaviour of DOLP with increasing SZA is consistent with the findings from stellar polarimetry (Bailey et al., 2008; Ulanowski et al., 2007b), although these studies are targeted on longer wavelengths and potential Rayleigh contamination my vary our results towards probing particle orientation."

10.       *"Line 438-440 "The observed excess in LP... (Figure 8b).*
*Can you reference this? This quite a speculation. How can you be sure that this LP vs AirMass relationship is related to particle orientation? Other considerations maybe be at play here such changes in atmospheric loading and size distribution, variability in dust composition or variability of multiple scattering through the day due to loading changes."*

**Author's Response:** We, again, deeply appreciate the reviewer for challenging our arguments and therefore strengthen the paper rationale as we incorporate the input. The relationship between the excess linear polarization and aerosols is a complex one and can be influenced by several factors other than particle orientation, such as: i. the particle size distribution and particle shape, ii. the dust particle composition as it can greatly affect their polarizing properties, iii. the observation geometry – including solar zenith angle (hence airmass) and iv. the particle concentration and spatial distribution. So, as the reviewer correctly points out there is a level of ambiguity when stating that the linearity can be attributed to particle viewing geometry or particle orientation. Having said that, no other hypotheses currently exist for the origin of dichroic polarization itself, apart from particle alignment. In this context, a linear relationship between airmass and LP is expected (see response to Comment 8) and is fully consistent with the alignment hypothesis. In order to be entirely conclusive, further comparison of the polarization measurements with synergistic retrievals of the particle microphysical properties (co-located CIMEL sunphotometer and lidar system) under the specific dust cases are scheduled in a future study, which again is a challenging topic as inherent biases of these instruments should also be considered. Nonetheless, we might be able to narrow down to the cases where there is not much load variability within the day and the layer is relatively homogeneous (for example, the May 2020 cases shown in the manuscript exhibit such a behavior through preliminary study). In order to properly consider the size distribution daytime variability, sophisticated in-situ measurements are needed to complement SolPol. These were available during the ESA Cal/Val ASKOS 2022 experiment where we also operated the instrument, but due to overcast conditions and extreme winds the co-location of both was difficult.

The scope of this initial research was to present a novel dataset of linear polarization observations of elevated dust layers with an instrument otherwise used in astropolarimetric studies, and attempt to interpret the measurements as indications of particle orientation, considering the previous work. As a future step we plan to perform a more complete testing, by acquiring observations of high-AOD pollution cases, where we know that the particles are spherical and thus produce no dichroic extinction.

Therefore, considering the reviewer's comment, to clarify our argument we have rewritten **Lines 438-441** as:

"We expect that for small AOD values ($< 0.1$) the linear polarization values will be within the noise threshold as derived from the clean days' behaviour, as is the case here and since there are no observed dust events with such small loads no increasing trends are exhibited (dark blue circles). As the optical depth increases, we observe upward trends in linear polarization for a specific SZA values, mainly above $40°$, intensifying in proportion to the AOD, with the highest values of DOLP recorded under heavier dust loads. This indicates that, as expected, dichroic extinction is enhanced due to the presence of a larger concentration of preferentially oriented dust particles for fixed viewing angles and, consequently, for constant airmass, as seen in the linear relation of Figure8b. Furthermore, the observed LP is linearly proportional to the airmass, which could be attributed to aligned particle viewing geometry for fixed AODs (since in the instrument frame of reference particles become more aligned for increasing SZA), or the amount of aligned particles along the line of sight, or both (Fig. 8b). This linear dependence is seen more clearly when we inspect the correlation between DOLP and the AOD corrected with airmass (slant optical depth) in Figure 9. The dependence

exhibits a threshold at AOD values of 0.1, higher values corresponding solely to the presence of dust. As we move to larger dust loads this trend contains cases of increasing linear polarization values even for smaller airmasses. The correlation strength could be further tested in future studies with dust loads close to one, as particle orientation could affect the geometric formalism for the derivation of the AOD."

11.      *"Line 510 : can you add a reference where this statement is explained? I mean why the mean orientation should be 60 degrees?"*

**Author's Response:** The specific statement regarding mean alignment angle for randomly oriented is erroneous and it was an unfortunate remnant of an early, pre-submission manuscript version. The 60° mean alignment angle refers to the mean polar angle and does not imply the existence of any form of preferred/dominant orientation (as expected from random alignment, by definition). The distribution of alignment angles measured around any polar axis, including the line of sight which is the axis relevant here, is uniform for randomly oriented particles, therefore the mean angle has no significance as an alignment angle. Thus, we have erased the sentence in question.